# Traveling Waves Encode the Recent Past and Enhance Sequence Learning

**T. Anderson Keller**[*]
The Kempner Institute for the Study
of Natural and Artificial Intelligence
Harvard University, USA

**Lyle Muller**
Department of Mathematics
Western University, CA

**Terrence Sejnowski**
Computational Neurobiology Lab
Salk Institute for Biological Studies, USA

**Max Welling**
Amsterdam Machine Learning Lab
University of Amsterdam, NL

## Abstract

Traveling waves of neural activity have been observed throughout the brain at a diversity of regions and scales; however, their precise computational role is still debated. One physically inspired hypothesis suggests that the cortical sheet may act like a wave-propagating system capable of invertibly storing a short-term memory of sequential stimuli through induced waves traveling across the cortical surface, and indeed many experimental results from neuroscience correlate wave activity with memory tasks. To date, however, the computational implications of this idea have remained hypothetical due to the lack of a simple recurrent neural network architecture capable of exhibiting such waves. In this work, we introduce a model to fill this gap, which we denote the Wave-RNN (wRNN), and demonstrate how such an architecture indeed efficiently encodes the recent past through a suite of synthetic memory tasks where wRNNs learn faster and reach significantly lower error than wave-free counterparts. We further explore the implications of this memory storage system on more complex sequence modeling tasks such as sequential image classification and find that wave-based models not only again outperform comparable wave-free RNNs while using significantly fewer parameters, but additionally perform comparably to more complex gated architectures such as LSTMs and GRUs.

## 1 Introduction

Since the earliest neural recordings (Caton, 1875; Beck, 1890), neural oscillations and the spatial organization of neural activity have persisted as topics of great interest in the neuroscience community. Consequently, a plethora of hypothesized potential functions for these widely observed phenomena have been put forth in the literature. For example: Pitts & McCulloch (1947) propose that alpha oscillations perform 'cortical scanning' akin to radar; Milner (1974) suggests that synchrony may serve to segregate the visual scene and 'bind' individual features into cohesive objects; (Engel et al., 2001) suggest brain-wide oscillations serve as 'top-down' stored knowledge and contextual influence for local processing; Raghavachari et al. (2001) show theta is consistent with a gating mechanism for human working memory; Buzsáki et al. (2013) posit that oscillations form a hierarchical system that offers a syntactical structure for spike traffic; and Liebe et al. (2012) & de Mooij-van Malsen et al. (2023) implicate oscillatory coherence with the transfer of information on working memory tasks.

Recently, the advents of high density multi-electrode arrays and high resolution imaging have led to the discovery that many of the oscillations observed in the brain are better described as traveling waves of activity rather than precise zero-lag synchrony (Muller et al., 2018). For example, alpha and theta oscillations have been measured to precisely correspond to traveling waves in both the cortex and hippocampus of humans (Lubenov & Siapas, 2009; Lozano-Soldevilla & VanRullen, 2019; Zhang et al., 2018). Furthermore, it has become increasingly clear that wave-like dynamics are

---

[*]Correspondence: T.Anderson.Keller@gmail.com, work completed while part of the UvA-Bosch Delta Lab.

prevalent throughout the brain, from local (Muller et al., 2014) to global (Muller et al., 2016) scales, and across virtually all brain regions measured (Townsend et al., 2015; Gu et al., 2021b; Pang et al., 2023). Research has correlated wave activity with a range of functional roles rivaling oscillations in diversity and number, including: perceptual awareness (Davis et al., 2020); attentional scanning (Fries, 2023); information transfer (Rubino et al., 2006); motor control sequencing and topographic organization (Takahashi et al., 2011); integrating visual sensory signals with saccadic eye movements (Zanos et al., 2015); and coordinating and reconfiguring distinct functional regions (Xu et al., 2023).

Most relevant to the present study, significant work has shown correlations between wave dynamics and memory. For example: King & Wyart (2021) provide evidence that traveling waves propagating from posterior to anterior cortex serve to encode sequential stimuli in an overlapping 'multiplexed' manner; Sauseng et al. (2002) show that traveling theta oscillations propagate from anterior to posterior regions during retrieval from long-term memory; and Zabeh et al. (2023) show traveling beta waves in frontal and parietal lobes encode memory of recent rewards. As described by Muller et al. (2018), one way to understand the relation between traveling waves and memory comes from an analogy to physical wave systems. Specifically, non-dissipative wave-propagating systems can be shown to contain all information about past disturbances in a 'time-reversible' manner. In contrast, in a static wave-free 'bump' system, onset time of stimuli are tied to position, and therefore information about the sequential order of multiple inputs at the same position is lost. This is visualized in Figure 1, and is demonstrated experimentally in Section 3. Although an imperfect analogy to the brain, such ideas have inspired neuroscientists to suggest that the cortical surface may act like such a wave-propagating system to efficiently encode recent sequential inputs as a form of working memory. In this work, we propose a complimentary mechanistic understanding of the relation between traveling waves and memory by suggesting that a wave-propagating hidden state can be thought of like a register or 'stack' to which inputs are sequentially written or 'pushed'. The propagation of waves then serves to prevent overwriting of past input, allowing for superior invertible memory storage.

In concert with these theories of potential function, an almost equal number of arguments have been put forth suggesting that traveling waves are 'epiphenomial' or merely the inert byproduct of more fundamental causal neural processes without any causal power of their own. One driving factor behind the continuation of this controversy is a lack of sufficiently capable models which permit the study of the computational role of waves in a task-relevant manner. In the computational neuroscience literature, there exist large-scale spiking neural network (SNN) models which exhibit traveling waves of activity (Davis et al., 2021); however, due to the computational complexity of SNNs, such models are still not feasible to train on real world tasks. These models therefore lack the ability to demonstrate the implications of traveling waves on memory. In the machine learning community, Keller & Welling (2023) recently introduced the Neural Wave Machine which exhibits traveling waves of activity in its hidden state in the service of sequence modeling; however, due to its construction as a network of coupled oscillators, it is impossible to disassociate the memory-enhancing contributions of oscillations (as described by Rusch & Mishra (2021a)) from those of traveling waves.

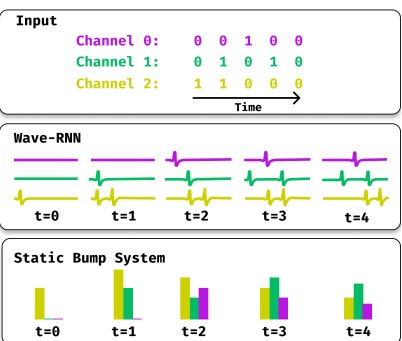

Figure 1: Three binary input signals (top), a corresponding wave-RNN hidden state (middle), and wave-free static bump system (bottom). At each timestep we are able decode both the onset time and channel of each input from the wave-RNN state. In the wave-free system, relative timing information is lost for inputs on the same channel, hindering learning and recall for sequential inputs.

In this work, we derive and introduce a simple non-oscillatory recurrent neural network (RNN) architecture which exhibits traveling waves in its hidden state and subsequently study the performance implications of these waves when compared directly with identical wave-free 'bump system' RNN counterparts. Ultimately, we find that wave-based models are able to solve sequence modeling tasks of significantly greater length, learn faster, and reach lower error than non-wave counterparts, approaching performance of more complicated state of the art architectures – thereby providing some of the first computational evidence for the benefits of wave-based memory systems. We therefore present our model as both a proof of concept of the computational role of wave-based memory systems, as well as a starting point for the continued development of such systems in the future.

## 2 TRAVELING WAVES IN RECURRENT NEURAL NETWORKS

In this section, we outline how to integrate traveling wave dynamics into a simple recurrent neural network architecture and provide preliminary analysis of the emergent waves.

**Simple Recurrent Neural Networks.** In order to reduce potential confounders in our analysis of the impact of waves on memory, we strive to study the simplest possible architecture which exhibits traveling waves in its hidden state. To accomplish this, we start with the simple recurrent neural network (sRNN) defined as follows. For an input sequence $\{\mathbf{x}_t\}_{t=0}^T$ with $\mathbf{x}_t \in \mathbb{R}^d$, and hidden state $\mathbf{h}_0 = \mathbf{0} \,\&\, \mathbf{h}_t \in \mathbb{R}^N$, an sRNN is defined with the following recurrence: $\mathbf{h}_{t+1} = \sigma(\mathbf{U}\mathbf{h}_t + \mathbf{V}\mathbf{x}_t + \mathbf{b})$ where the input encoder and recurrent connections are both linear, i.e. $\mathbf{V} \in \mathbb{R}^{N \times d}$ and $\mathbf{U} \in \mathbb{R}^{N \times N}$, where $N$ is the hidden state dimensionality and $\sigma$ is a nonlinearity. The output of the network is then given by another linear map of the final hidden state: $\mathbf{y} = \mathbf{W}\mathbf{h}_T$, with $\mathbf{W} \in \mathbb{R}^{o \times N}$.

**Wave-Recurrent Neural Networks.** To integrate wave dynamics into the sRNN, we start with the simplest equation which encapsulates our goal, the one-dimensional one-way wave equation:

$$\frac{\partial h(x, t)}{\partial t} = \nu \frac{\partial h(x, t)}{\partial x} \tag{1}$$

Where $t$ is our time coordinate, $x$ defines the continuous spatial coordinate of our hidden state, and $\nu$ is the wave velocity. We can see that if we discretize this equation over space and time, with timestep $\Delta t$, defining $h(x, t) = h_t^x$ as the activation of the $x$'th neuron at timestep $t$, this is equivalent to multiplication of the hidden state vector with the following circulant matrix:

$$\mathbf{h}_{t+1} = \Sigma \mathbf{h}_t \quad \text{where} \quad \Sigma = \begin{bmatrix} 1 - \nu' & \nu' & 0 & \cdots & 0 \\ 0 & 1 - \nu' & \nu' & \cdots & 0 \\ \vdots & \vdots & \vdots & \ddots & \vdots \\ 0 & 0 & 0 & \cdots & \nu' \\ \nu' & 0 & 0 & \cdots & 1 - \nu' \end{bmatrix}. \tag{2}$$

where $\nu' = \nu \Delta t$. A common linear operator which has a similar circulant structure to $\Sigma$ is convolution. Specifically, assuming a single channel length 3 convolutional kernel $\mathbf{u} = [0, 1 - \nu, \nu]$, we see the following equivalence: $\mathbf{u} \star \mathbf{h}_{t-1} = \Sigma \mathbf{h}_{t-1}$, where $\star$ defines circular convolution over the hidden state dimensions $N$. Intuitively this can be thought of as a ring of neurons with shared recurrent local connectivity. We therefore propose to define the Wave-RNN (wRNN) with the following recurrence:

$$\mathbf{h}_{t+1} = \sigma(\mathbf{u} \star \mathbf{h}_t + \mathbf{V}\mathbf{x}_t + \mathbf{b}) \tag{3}$$

In practice we find that increasing the number of channels helps the model to learn significantly faster and reach lower error. To do this, we define $\mathbf{u} \in \mathbb{R}^{c \times c \times f}$ where $c$ is the number of channels, and $f$ is the kernel size, and we reshape the hidden state from a single $N$ dimensional circle to $c$ separate $n = \lfloor \frac{N}{c} \rfloor$ dimensional circular channels (e.g. $\mathbf{h} \in \mathbb{R}^{c \times n}$). For the activation $\sigma$, we follow recent work which finds that linear and rectified linear activations have theoretical and empirical benefits for long-sequence modeling (Le et al., 2015; Orvieto et al., 2023). Finally, similar to the prior work with recurrent neural networks (Gu et al., 2022), we find careful initialization can be crucial for the model to converge more quickly and reach lower final error. Specifically, we initialize the convolution kernel such that the matrix form of the convolution is exactly that of the shift matrix $\Sigma$ for each channel separately, with $\nu = 1$. Furthermore, we find that initializing the matrix $\mathbf{V}$ to be all zero except for a single identity mapping from the input to a single hidden unit to further drastically improve training speed. Intuitively, these initalizations combined can be seen to support a separate traveling wave of activity in each channel, driven by the input at a single source location. Pseudocode detailing these initalizations can be found in Appendix B, and an ablation study can be found in Table 4.

**Baselines.** In order to isolate the effect of traveling waves on model performance, we desire to pick baseline models which are as similar to the Wave-RNN as possible while not exhibiting traveling waves in their hidden state. To accomplish this, we rely on the Identity Recurrent Neural Network (iRNN) of Le et al. (2015). This model is nearly identical to the Wave-RNN, constructed as a simple RNN with $\sigma = \text{ReLU}$, but uses an identity initialization for $\mathbf{U}$. Due to this initialization, the iRNN can be seen to be nearly analagous to the 'bump system' described in Figure 1, where activity does not propagate between neurons but rather stays as a static/decaying 'bump'. Despite its simplicity the iRNN is found to be comparable to LSTM networks on standard benchmarks, and thus represents the

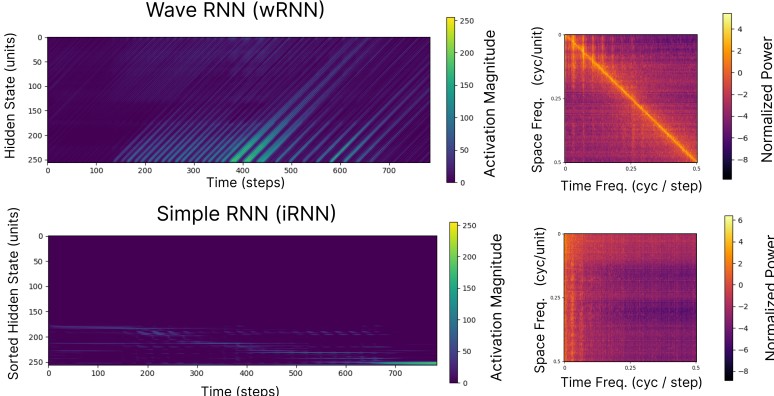

Figure 2: Visualization of hidden state (left) and associated 2D Fourier transform (right) for a wRNN (top) and iRNN (bottom) after training on the sMNIST task. We see the wRNN exhibits a clear flow of activity across the hidden state (diagonal bands) while the iRNN does not. Similarly, from the 2D space-time fourier transform, we see the wRNN exhibits significantly higher power along the diagonal corresponding to the wave propagation velocity of 1 unit/step (Mahmoud et al., 1988).

ideal highly capable simple recurrent neural network which is comparable to the Wave-RNN. We note that, as in the original work, we allow all parameters of the matrix $\mathbf{U}$ to be optimized.

**Visualization of Traveling Waves.** Before we study the memory capabilities of the wRNN, we first demonstrate that the model does indeed produce traveling waves within its hidden state. To do this, in Figure 2, for the best performing (wRNN & iRNN) models on the Sequential MNIST task of the proceeding section, we plot in the top row the activations of our neurons (vertical axis) over time (horizontal axis) as the RNNs process a sequence of inputs (MNIST pixels). As can be seen, there are distinct diagonal bands of activation for the Wave-RNN (left), corresponding to waves of activity propagating between hidden neurons over time. For the baseline simple RNN (iRNN) right, despite sorting the hidden state neurons by 'onset time' of maximum activation to uncover any potential traveling wave activity, we see no such bands exist, but instead stationary bumps of activity exist for a duration of time and then fade. In the bottom row, following the analysis techniques of Davis et al. (2021), we plot the corresponding 2D Fourier transform of the above activation time series. In this plot, the vertical axis corresponds to spatial frequencies while the horizontal axis corresponds to temporal frequencies. In such a 2D frequency space, a constant speed traveling wave (or general moving object (Mahmoud et al., 1988)) will appear as a linear correlation between space and time frequencies where the slope corresponds to the speed. Indeed, for the Wave-RNN, we see a strong band of energy along the diagonal corresponding to our traveling waves with velocity $\approx 1 \frac{\text{unit}}{\text{timestep}}$; as expected, for the iRNN we see no such diagonal band in frequency space. In Appendix C Figure 10 we show additional visualizations of waves propagating at multiple different speeds simultaneously within the same network, demonstrating flexibility of learned wave dynamics. Further, in Appendix C Figure 18, we show how the wave dynamics change through training for a variety of different initalizations and architectures. We see that even in wRNN models with random recurrent kernel $\mathbf{u}$ initalizations, although waves are not present at initialization, they are learned through training in the service of sequence modeling, indicating they are a valuable solution for memory tasks.

## 3 EXPERIMENTS

In this section we aim to leverage the model introduced in Section 2 to test the computational hypothesis that traveling waves may serve as a mechanism to encode the recent past in a wave-field short-term memory. To do this, we first leverage a suite of frequently used synthetic memory tasks designed to precisely measure the ability of sequence models to store information and learn dependencies over variable length timescales. Following this, we use a suite of standard sequence modeling benchmarks to measure if the demonstrated short-term memory benefits of wRNNs persist in a more complex regime. For each task we perform a grid search over learning rates, learning rate schedules, and gradient clip magnitudes, presenting the best performing models from each category on a held-out validation set in the figures and tables. In Appendix B we include the full ranges of each grid search as well as exact hyperparameters for the best performing models in each category.

**Copy Task.** As a first analysis of the impact of traveling waves on memory encoding, we measure the performance of the wRNN on the standard 'copy task', as frequently employed in prior work (Graves et al., 2014; Arjovsky et al., 2016; Gu et al., 2020b). The task is constructed of sequences of categorical inputs of length $T + 20$ where the first 10 elements are randomly chosen one-hot vectors representing a category in $\{1, \ldots 8\}$. The following $T$ tokens are set to category 0, and form the time duration where the network must hold the information in memory. The next token is set to 9, representing a delimiter, signaling the RNN to begin reproducing the stored memory as output, and the final 9 tokens are again set to category 0. The target for this task is another categorical sequence of length $T + 20$ with all elements set to category 0 except for the last 10 elements containing the initial random sequence of the input to be reproduced. At a high level, this task tests the ability for a network to encode categorical information and maintain it in memory for $T$ timesteps before eventually reproducing it. Given the hypothesis that traveling waves may serve to encode information in an effective 'register', we hypothesize that wave-RNNs should perform significantly better on this task than the standard RNN. For each sequence length we compare wRNNs with 100 hidden units per channel and 6 channels ($n = 100, c = 6$) with two baselines: iRNNs of comparable parameter counts ($n = 100 \Rightarrow$ 12k params.), and iRNNs with comparable numbers of activations/neurons ($n = 625$) but a significantly greater parameter count ($\Rightarrow$ 403k params.). In Figure 3, we show the performance of the best iRNNs and wRNNs, obtained from our grid search, for $T = \{0, 30, 80\}$. We see that the wRNNs achieve more than 5 orders of magnitude lower loss and learn exponentially faster for all sequence lengths. Furthermore, we see that the iRNN with ($n = 625$) still fails to perform even close to the wRNN despite having an equivalent number of activations and nearly 40 times more parameters. From the visualization of the model outputs in Figure 4, we see that the iRNN has trouble holding items in memory for longer than 10 timesteps, while the comparable wRNN has no problem copying data for up to 500 timesteps. Ultimately, this experiment demonstrates exactly the distinction between traveling wave and static systems illustrated in Figure 1 – the iRNN (static) system is unable to accurately maintain the relative order of sequence elements that have the same input encoding, while the wRNN wave field has no problem encoding both timing and position, facilitating decoding.

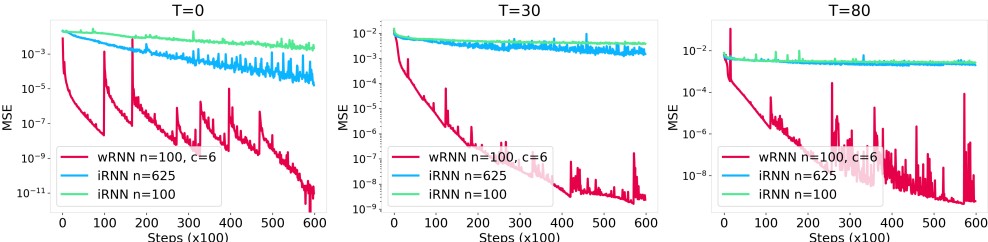

Figure 3: Copy task with lengths T={0, 30, 80}. wRNNs achieve $> 5$ orders of magnitude lower loss than iRNNs with approximately equal number of parameters ($n = 100$) and activations ($n = 625$).

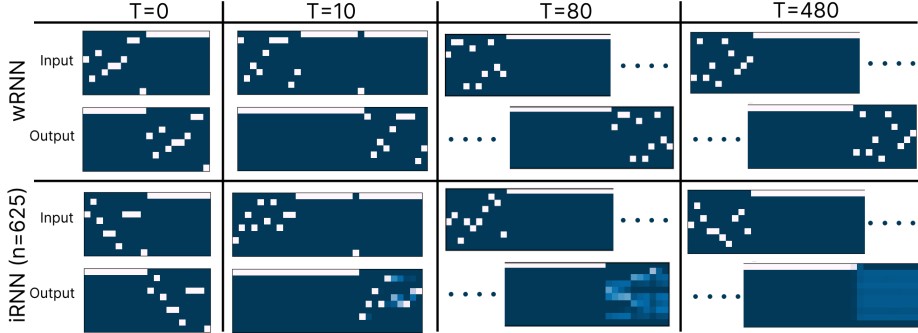

Figure 4: Examples from the copy task for wRNN (n=100, c=6) and iRNN (n=625). We see the iRNN loses significant accuracy after T=10 while the wRNN remains perfect at T=480 (MSE $\approx 10^{-9}$).

**Adding Task.** To bolster our findings from the copy task, we employ the long-sequence addition task originally introduced by Hochreiter & Schmidhuber (1997). The task consists of a two dimensional input sequence of length $T$, where the first dimension is a random sample from $\mathcal{U}([0, 1])$, and the second dimension contains only two non-zero elements (set to 1) in the first and second halves of the sequence respectively. The target is the sum of the two elements in the first dimension which

correspond to the non-zero indicators in the second dimension. Similar to the copy task, this task allows us to vary the sequence length and measure the limits of each model's ability. The original iRNN paper (Le et al., 2015) demonstrated that standard RNNs without identity initialization struggle to solve sequences with $T > 150$, while the iRNN is able to perform equally as well as an LSTM, but begins to struggle with sequences of length greater than 400 (a result which we reconfirm here). In our experiments depicted in Figure 5 and Table 1, we find that the wRNN not only solves the task much more quickly than the iRNN, but it is also able solve significantly longer sequences than the iRNN (up to 1000 steps). In these experiments we use an iRNN with $n = 100$ hidden units (10.3k parameters) and a wRNN with $n = 100$ hidden units and $c = 27$ channels (10.29k parameters).

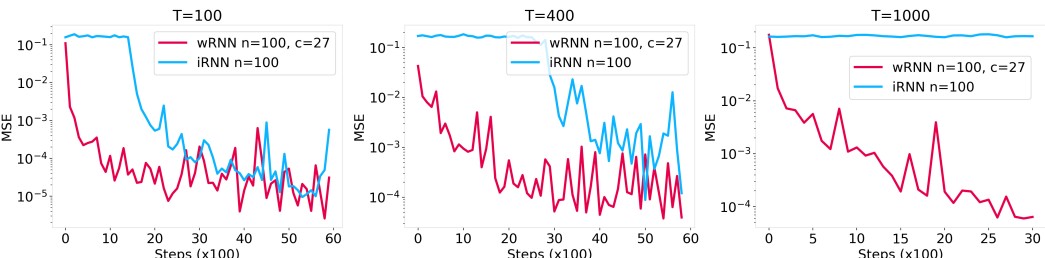

Figure 5: wRNN and iRNN Training curves on the addition task for three different sequence lengths (100, 400, 1000). We see that the wRNN converges significantly faster than the iRNN on all lengths, achieves lower error, and can solve tasks which are significantly longer.

| | Seq. Length (T) | 100 | 200 | 400 | 700 | 1000 |
|---|---|---|---|---|---|---|
| iRNN | Test MSE | $1 \times 10^{-5}$ | $4 \times 10^{-5}$ | $1 \times 10^{-4}$ | 0.16 | 0.16 |
| | Solved Iter | 14k | 22k | 30k | × | × |
| wRNN | Test MSE | $\mathbf{4 \times 10^{-6}}$ | $\mathbf{2 \times 10^{-5}}$ | $\mathbf{4 \times 10^{-5}}$ | $\mathbf{8 \times 10^{-5}}$ | $\mathbf{6 \times 10^{-5}}$ |
| | Solved Iter. | **300** | **1k** | **1k** | **3k** | **2k** |

Table 1: Long sequence addition task for different sequence lengths. The wRNN finds the task solution (defined as MSE $\leq 5 \times 10^{-2}$) multiple orders of magnitude quicker and is able to solve much longer tasks than the iRNN. The × indicates the model never solved the task after 60k iterations.

**Sequential Image Classification.** Given the dramatic benefits that traveling waves appear to afford in the synthetic memory-specific tasks, in this section we additionally strive to measure if waves will have any similar benefits for more complex sequence tasks relevant to the machine learning community. One common task for evaluating sequence models is sequential pixel-by-pixel image classification. In this work we specifically experiment with three sequential image tasks: sequential MNIST (sMNIST), permuted sequential MNIST (psMNIST), and noisy sequential CIFAR10 (nsCIFAR10). The MNSIT tasks are constructed by feeding the 784 pixels of each image of the MNIST dataset one at a time to the RNN, and attempting to classify the digit from the hidden

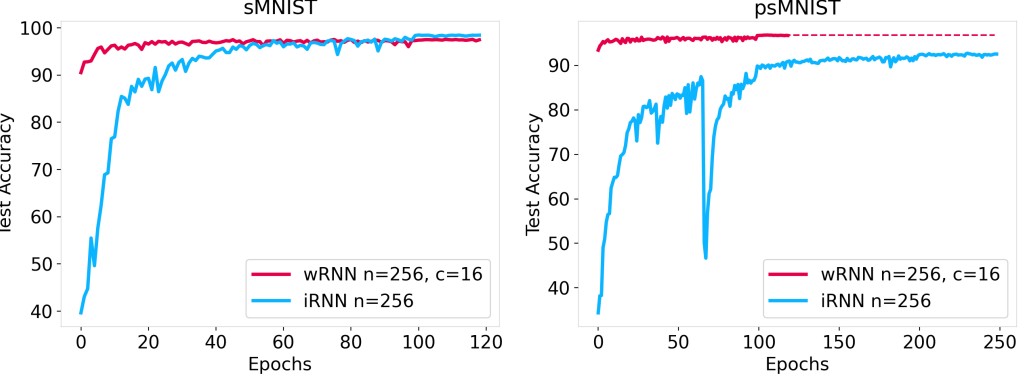

Figure 6: sMNIST (left) and psMNIST (right) training curves for the iRNN & wRNN. The wRNN trains much faster and is virtually unaffected by the sequence permutation, while the iRNN suffers.

state after the final timestep. The permuted variant applies a random fixed permutation to the order of the pixels before training, thereby increasing the task difficulty by preventing the model from leveraging statistical correlations between nearby pixels. The nsCIFAR10 task is constructed by feeding each row of the image ($32 \times 3$ pixels) flattened as vector input to the network at each timestep. This presents a significantly higher input-dimensionality than the MNIST tasks, and additionally contains more complicated sequence dependencies due to the more complex images. To further increase the difficulty of the task, the sequence length is padded from the original length (32), to a length of 1000 with random noise. Therefore, the task of the model is not only to integrate the information from the original 32 sequence elements, but additionally ignore the remaining noise elements. As in the synthetic tasks, we again perform a grid search over learning rates, learning rate schedules, and gradient clip magnitudes. Because of our significant tuning efforts, we find that our baseline iRNN results are significantly higher than those presented in the original work (98.5% vs. 97% on sMNIST, 91% vs. 81% on psMNIST), and additionally sometimes higher than many 'state of the art' methods published after the original iRNN. In the tables below we indicate results from the original work by a citation next to the model name, and lightly shade the rows of our results.

In Table 2, we show our results in comparison with existing work on the sMNIST and psMNIST. Despite the simplicity of our proposed approach, we see that it performs favorably with many carefully crafted RNN and convolutional architectures. We additionally include wRNN + MLP, which is the same as the existing wRNN, but replaces the output map **W** with a 2-layer MLP. We see this increases performance significantly, suggesting the linear decoder of the basic wRNN may be a performance bottleneck. In Figure 6 (left), we plot the training accuracy of the best performing wRNN compared with the best performing iRNN over training iterations on the sMNIST dataset. We see that while the iRNN reaches a slightly higher final accuracy (+0.9%), the wRNN trains remarkably faster at the beginning

| Model | sMNIST | psMNIST | $n$ / $\#\theta$ |
|---|---|---|---|
| uRNN[1] | 95.1 | 91.4 | 512 / 9k |
| iRNN | 98.5 | 92.5 | 256 / 68k |
| LSTM[2] | 98.8 | 92.9 | 256 / 267k |
| GRU[2] | 99.1 | 94.1 | 256 / 201k |
| NWM[8] | 98.6 | 94.8 | 128 / 50k |
| IndRNN (6L)[3] | 99.0 | 96.0 | 128 / 83k |
| Lip. RNN[6] | 99.4 | 96.3 | 128 / 34k |
| coRNN[7] | 99.3 | 96.6 | 128 / 34k |
| LEM[2] | 99.5 | 96.6 | 128 / 68k |
| wRNN (16c) | 97.6 | 96.7 | 256 / 47k |
| URLSTM[4] | 99.2 | 97.6 | 1024 / 4.5M |
| wRNN + MLP | 97.5 | 97.6 | 256 / 420k |
| FlexTCN[5] | 99.6 | 98.6 | - / 375k |

Table 2: sMNIST & psMNIST test accuracy, sorted by psMNIST score. Baseline values from: [1] Arjovsky et al. (2016), [2] Rusch et al. (2022), [3] Li et al. (2018), [4] Gu et al. (2020b), [5] Romero et al. (2022), [6] Erichson et al. (2021), [7] Rusch & Mishra (2021a), [8] Keller & Welling (2023).

of training, taking the iRNN roughly 50 epochs to catch up. On the right of the figure, we plot the models' performance on the permuted variant of the task (psMNIST) and see the performance of the Wave-RNN is virtually unaffected, while the simple RNN baseline suffers dramatically. Intuitively, this performance difference on the permuted task may be seen to come from the fact that by directly encoding the input into the wave-field 'buffer', the wRNN is able to learn to classify sequences invariant of the ordering or permutations of the input (through the fully-connected readout matrix **W**), while the iRNN has no such buffer and thus struggles to encode sequences with less temporal structure.

We note that in addition to faster training and higher accuracy, the wRNN model additionally exhibits substantially greater parameter efficiency than the iRNN due to its convolutional recurrent connections in place of fully connected layers. To exemplify this, in Figure 7 we show the accuracy (y-axis) of a suite of wRNN models plotted as a function of the number of parameters (x-axis). We see that compared with the iRNN, the wRNN reaches near maximal performance with significantly fewer parameters, and retains a performance gap over the iRNN with increased parameter counts.

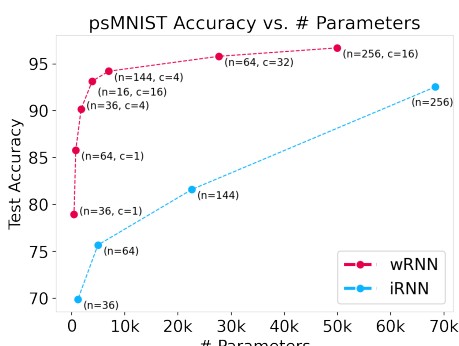

Figure 7: Num. parameters vs. accuracy for wRNNs & iRNNs on psMNIST.

Finally, to see if the benefits of the wRNN extend to more complicated images, we explore the noisy sequential CIFAR10 task. In Figure 8 we plot the training curves of the best performing models on this dataset, and see that the Wave-RNN still maintains a significant advantage over the iRNN in this setting. In Table 3, we see the performance of the wRNN is ahead of standard gated architectures such as GRUs and LSTMs, but also ahead of more recent complex gated architectures such as the Gated anti-symmetric RNN (Chang et al., 2019). We believe that these results therefore serve as strong evidence in support of the hypothesis that traveling waves may be a valuable inductive bias for encoding the recent past and thereby facilitate long-sequence learning.

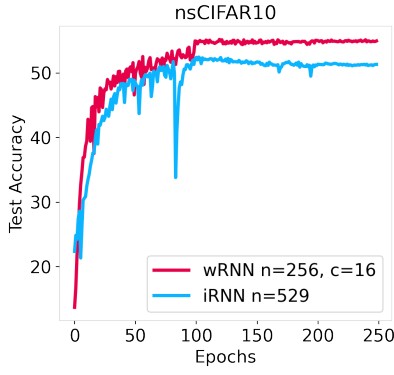

| Model | Acc. | # units / params |
|---|---|---|
| LSTM [1] | 11.6 | 128 / 116k |
| GRU [1] | 43.8 | 128 / 88k |
| anti-sym. RNN [2] | 48.3 | 256 / 36k |
| iRNN | 51.3 | 529 / 336k |
| Incremental RNN [3] | 54.5 | 128 / 12k |
| Gated anti-sym. RNN [2] | 54.7 | 256 / 37k |
| wRNN (16c) | 55.0 | 256 / 435k |
| Lipschits RNN [4] | 57.4 | 128 / 46k |
| coRNN [5] | 59.0 | 128 / 46k |
| LEM [1] | 60.5 | 128 / 116k |

Figure 8: Training curves on noisy sequential CIFAR10. We again see the wRNN trains faster and reaches higher accuracy than the wave-free model.

Table 3: Test set accuracy on the noisy sequential CIFAR dataset sorted by performance. Baseline values from: [1]Rusch et al. (2022), [2]Chang et al. (2019), [3] Kag et al. (2020), [4]Erichson et al. (2021), [5]Rusch & Mishra (2021a)

**Ablation Experiments.** Finally, we include ablation experiments to validate the architecture choices for the Wave-RNN. For each of the results reported below, we again grid search over learning rates, activation functions, initalizations, and gradient clipping values. In Table 4, we show the performance of the wRNN on the copy task as we ablate various proposed components such as convolution, $\mathbf{u}$-shift initialization, and $\mathbf{V}$ initialization (as described in Section 2). At a high level, we see that the wRNN as proposed performs best, with $\mathbf{u}$-shift initialization having the biggest impact on performance, allowing the model to successfully solve tasks greater than length $T = 10$. In addition to ablating the wRNN, we additionally explore initializing the iRNN with a shift initialization ($\mathbf{U} = \Sigma$) and sparse identity initialization for $\mathbf{V}$ to disassociate these effects from the effect of the convolution operation. We see that the addition of $\Sigma$ initialization to the iRNN improves its performance dramatically, but it never reaches the same level of performance of the wRNN – indicating that the sparsity and tied weights of the convolution operation are critical to memory storage and retrieval on this task.

| Model | Sequence Length (T) | | | |
|---|---|---|---|---|
| | 0 | 10 | 30 | 80 |
| wRNN | $\mathbf{9 \times 10^{-12}}$ | $1 \times 10^{-10}$ | $\mathbf{8 \times 10^{-11}}$ | $\mathbf{1 \times 8^{-11}}$ |
| - $\mathbf{V}$-init | $\underline{1 \times 10^{-11}}$ | $\mathbf{2 \times 10^{-11}}$ | $4 \times 10^{-10}$ | $\underline{4 \times 10^{-11}}$ |
| - $\mathbf{u}$-shift-init | $5 \times 10^{-11}$ | $3 \times 10^{-10}$ | $7 \times 10^{-4}$ | $6 \times 10^{-4}$ |
| - $\mathbf{V}$-init - $\mathbf{u}$-shift-init | $8 \times 10^{-10}$ | $1 \times 10^{-4}$ | $3 \times 10^{-4}$ | $7 \times 10^{-4}$ |
| iRNN (n=100) | $1 \times 10^{-4}$ | $3 \times 10^{-3}$ | $2 \times 10^{-3}$ | $1 \times 10^{-3}$ |
| + $\Sigma$-init | $1 \times 10^{-8}$ | $1 \times 10^{-7}$ | $2 \times 10^{-7}$ | $2 \times 10^{-5}$ |
| + $\Sigma$-init + $\mathbf{V}$-init | $1 \times 10^{-7}$ | $1 \times 10^{-7}$ | $1 \times 10^{-6}$ | $8 \times 10^{-6}$ |

Table 4: Ablation test results (MSE) on the copy task. Best results are bold, second best underlined.

## 4 DISCUSSION

In this work we have discussed theories from neuroscience relating traveling waves to working memory, and have provided one of the first points of computational evidence in support of these

theories through experiments with our Wave-RNN – a novel minimal recurrent neural network capable of exhibiting traveling waves in its hidden state. In the discussion below we include a brief summary of related work, limitations, and future work, with an extended survey of related work in Appendix A.

**Related Work.** As mentioned in the introduction, most related to this work in practice, Keller & Welling (2023) showed that an RNN parameterized as a network of locally coupled oscillators (NWM) similarly exhibits traveling waves in its hidden state, and that these waves served as a bias towards learning structured representations. This present paper can be seen as a reduction of the Keller & Welling (2023) model to its essential elements, allowing for a controlled study of hypotheses relating traveling waves and memory. Additionally, we see in practice that this reduction sometimes improves performance, as on the psMNIST task in Table 2, where the wRNN outperforms the NWM.

Another related line of work proposes to solve long-sequence memory tasks by taking inspiration for biological 'time-cells' (Jacques et al., 2021; 2022). The authors use temporal convolution with a set of exponentially scaled fixed kernels to extract a set of time-scale-invariant features at each layer. Compared with the wRNN, the primary conceptual difference is the logarithmic scaling of time features which is purported to increase efficiency for extremely long time dependencies. As we demonstrate in Appendix Fig. 10, however, the wRNN is also capable of learning waves of multiple speeds, and therefore with proper initialization may also exhibit a logarithmic scaling of time-scales.

Similar in results to our work, Chen et al. (2022) demonstrated that in a predictive RNN autoencoder learning sequences, Toeplitz connectivity emerges spontaneously, replicating multiple canonical neuroscientific measurements such as one-shot learning and place cell phase precession. Our results in Figure 18 further support these findings that with proper training and connectivity constraints, recurrent neural networks can learn to exhibit traveling wave activity in service of solving a task.

Finally, we note that the long-sequence modeling goal of this work can be seen as similar to that of the recently popularized Structured State Space Models (S4, HIPPO, LSSL) (Gu et al., 2022; 2020a; 2021a). In practice the recurrent connectivity matrix of the wRNN is quite different than the HiPPO initialization pioneered by these works, however we do believe that linear recurrence may be an equivalently useful property for wRNNs. Importantly, however, the H3 mechanism of Fu et al. (2023) does indeed include a 'shift SSM' with a toeplitz structure very similar to that of the wRNN. However, this shift initialization was subsequently dropped in following works such as Heyena and Mamba (Poli et al., 2023; Gu & Dao, 2023). In future work we intend to explore this middle ground between the wRNN and state space models as we find it to likely be fruitful.

**Limitations & Future Work.** The experiments and results in this paper are limited by the relatively small scale of the models studied. For example, nearly all models in this work rely on linear encoders and decoders, and consist of a single RNN layer. Compared with state of the art models (such as S4) consisting of more complex deep RNNs with skip connections and regularization, our work is therefore potentially leaving significant performance on the table. However, as described above, beginning with small scale experiments on standard architectures yields alternative benefits including more accurate hyperparameter tuning (due to a smaller search space) and potentially greater generality of conclusions. As the primary goal of this paper was to demonstrate the computational advantage of traveling waves over wave-free counterparts on memory tasks, the tradeoff for small scale was both necessary and beneficial. In future work, we plan to test the full implications of our results for the machine learning community and integrate the core concepts from the wRNN into more modern sequence learning algorithms, such as those used for language modeling.

We note that the parameter count for the wRNN on the CIFAR10 task is significantly higher than the other models listed in the table. This is primarily due to the linear encoder mapping from the high dimensionality of the input (96) to the large hidden state. In fact, for this model, the encoder $\mathbf{V}$ alone accounts for $> 90\%$ of the parameters of the full model (393k/435k). If one were to replace this encoder with a more parameter efficient encoder, such as a convolutional neural network or a sparse matrix (inspired by the initialization for $\mathbf{V}$), the model would thus have significantly fewer parameters, making it again comparable to state of the art. We leave this addition to future work, but believe it to be one of the most promising approaches to improving the wRNN's general competitiveness.

Finally, we believe this work opens the door for significant future work in the domains of theoretical and computational neuroscience. For example, direct comparison of the wave properties of our model with neurological recordings (such as those from (King & Wyart, 2021)) may provide novel insights into the mechanisms and role of traveling waves in the brain, analogous to how comparison

of visual stream recordings with convolutional neural network activations has yielded insights into the biological visual system (Cadieu et al., 2014).

## ACKNOWLEDGMENTS AND DISCLOSURE OF FUNDING

We would like to thank the creators of Weight & Biases (Biewald, 2020) and PyTorch (Paszke et al., 2019). Without these tools our work would not have been possible. We thank the Bosch Center for Artificial Intelligence for funding T. Anderson Keller for the initial stages of this project, and The Kempner Institute for funding him through the final stages of the project.

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

# Appendices

## Table of Contents

## A  RELATED WORK

Deep neural network architectures that exhibit some brain-like properties are related to ours: CORnet (Kubilius et al., 2019), a shallow convolutional network with added layer-wise recurrence shown to be a better match to primate neural responses; models with topographic organization, such as the TVAE (Keller & Welling, 2021b; Keller et al., 2021; Keller & Welling, 2021a) and TDANN (Lee et al., 2020); and models of hippocampal-cortex interactions such as the PredRAE (Chen et al., 2022), and the TEM (Whittington et al., 2020). Our work is unique in this space in that it is specifically focused on generating spatio-temporally synchronous activity, unlike prior work. Furthermore, we believe that our findings and approach may be complimentary to existing models, increasing their ability to model neural dynamics by inclusion of the Wave-RNN fundamentals such as locally recurrent connections and shift initalizations.

In the machine learning literature, there are a number of works which have experimented with local connectivity in recurrent neural networks. Some of the earliest examples include Neural GPUs (Łukasz Kaiser & Sutskever, 2016) and Convolutional LSTM Networks (Shi et al., 2015). These works found that using convolutional recurrent connections could be beneficial for learning algorithms and spatial sequence modeling respectively. Unlike these works however, the present paper explicitly focuses on the emergence of wave-like dynamics in the hidden state, and further studies how these dynamics impact computation. To accomplish this, we also focus on simple recurrent neural networks as opposed to the gated architectures of prior work – providing a less obfuscated signal as to the computational role of wave-like dynamics.

One line of work that is highly related in terms of application is the suite of models developed to increase the ability of recurrent neural networks to learn long time dependencies. This includes models such as Unitary RNNs (Arjovsky et al., 2016), Orthogonal RNNs (Henaff et al., 2016), expRNNs (Lezcano-Casado & Martínez-Rubio, 2019), the chronoLSTM (Tallec & Ollivier, 2018), anti.symmetric RNNs (Chang et al., 2019), Lipschitz RNNs (Erichson et al., 2021), coRNNs (Rusch & Mishra, 2021a), unicoRNNs (Rusch & Mishra, 2021b), LEMs (Rusch et al., 2022), periodic & quasi-periodic continous attractors (Park et al., 2023), Recurrent Linear Units (Orvieto et al., 2023), and Structured State Space Models (S4) (Gu et al., 2022). Additional models with external memory may also be considered in this category such as Neural Turing Machines (Graves et al., 2014), the DNC (Graves et al., 2016), memory augmented neural networks (Santoro et al., 2016) and Fast-weight RNNs (Ba et al., 2016). Although we leverage many of the benchmarks and synthetic tasks from these works in order to test our model, we note that our work is not intended to compete with state of the art on the tasks and thus we do not compare directly with all of the above models. Instead, the present papers intends to perform a rigorous empirical study of the computational implications of traveling waves in RNNs. To best perform this analysis, we find it most beneficial to compare directly with as similar of a model as possible which does not exhibit traveling waves, namely the iRNN. We do highlight, however, that despite being distinct from aforementioned models algorithmically, and arguably significantly simpler in terms of concept and implementation, the wRNN achieves highly competitive results. Finally, distinct from much of this prior work (except for the coRNN (Rusch & Mishra, 2021a) and DeepSITH Jacques et al. (2021)), our work uniquely leverages neuroscientific inspiration to solve the long-sequence memory problem, thereby potentially offering insights into neuroscience observations in return. Of the above mentioned models, our model is perhaps most intimately related in theory to the periodic and quasi-periodic attractor networks of Park et al. (2023),

and we therefore encourage readers to consult that work for a greater theoretical understanding of the wave-RNN.

In a final related paper, Benigno et al. (2022) studied a complex-valued recurrent neural network with distance-dependant connectivity and signal-propagation time-delay, in order to understand the potential computational role for traveling waves that have been observed in visual cortex. They showed that when recurrent strengths are set optimally, the network is able to perform long-term closed-loop video forecasting significantly better than networks lacking this spatially-organized recurrence. Our model is complimentary to this approach, focusing instead on the sequence integration capabilities of waves, rather than on forecasting, and leveraging a more traditional deep learning architecture.

## B  EXPERIMENT DETAILS

In this section, we include all experiment details including the grid search ranges and the best performing parameters. All code for reproducing the results can be found at the following repository: https://github.com/akandykeller/Wave_RNNs. The code was based on the original code base from the coRNN paper (Rusch & Mishra, 2021a) found at https://github.com/tk-rusch/coRNN.

**Pseudocode.**    Below we include an example implementation of the wRNN cell in Pytorch (Paszke et al., 2019):

```python
import torch.nn as nn

class wRNN_Cell(nn.Module):
    def __init__(self, n_in, n, c, k=3):
        super(RNN_Cell, self).__init__()
        self.n = n
        self.c = c
        self.V = nn.Linear(n_in, n * c)
        self.U = nn.Conv1d(c, c, k, 1, k//2, padding_mode='circular')
        self.act = nn.ReLU()

        # Sparse identity initialization for V
        nn.init.zeros_(self.V.weight)
        nn.init.zeros_(self.V.bias)
        with torch.no_grad():
            w = self.V.weight.view(c, n, n_in)
            w[:, 0] = 1.0

        # Shift initialization for U
        wts = torch.zeros(c, c, k)
        nn.init.dirac_(wts)
        wts = torch.roll(wts, 1, -1)
        with torch.no_grad():
            self.U.weight.copy_(wts)

    def forward(self, x, hy):
        hy = self.act(self.Wx(x).view(-1, self.c, self.n) + self.Wy(hy))
        return hy
```

**Figure 2.**    The results displayed in Figure 2 are from the best performing models of the sMNIST experiments, precisely the same as those reported in Figure 6 (left) and Table 2. The hyperparamters of these models are described in the **sMNIST** section below. To compute the 2D Fourier transform, we follow the procedure of Davis et al. (2021): we compute the real valued magnitude of the 2-dimensional Fourier transform of the hidden state activations over time (using `torch.fft.fft2(seq).abs()`). To account for the significant autocorrelation in the data, we normalize the output by the power spectrum of the spatially and temporally shuffled sequence of activations. We note that although this makes the diagonal bands more prominent, they are still clearly visible in the un-normalized spectrum. Finally, we plot the logarithm of the power for clarity.

**Copy Task.**    We construct each batch for the copy task as follows:

```python
def get_copy_task_batch(bsz, T):
    X = np.zeros((bsz, T+10))
    data = np.random.randint(low=1, high=9, size=(bsz, 10))
    X[:, :10] = data
    X[:, -(10+1)] = 9
    Y = np.zeros((bsz, T+10))
    Y[:, -10:] = X[:, :10]

    X = F.one_hot(torch.tensor(X, dtype=torch.int64).permute(1,0), 10)
    Y = torch.tensor(Y).int().permute(1,0)

    return X, Y
```

For the copy task, we train each model with a batch size of 128 for 60,000 batches. The models are trained with cross entropy loss, and optimized with the Adam optimizer. We grid search over the following hyperparameters for each model type:

- iRNN
  - Gradient Clip Magnitude: $[0, 1.0, 10.0]$
  - U Initialization: $[\mathbf{I}, \mathcal{U}(-\frac{1}{\sqrt{n}}, \frac{1}{\sqrt{n}})]$
  - Learning Rate: $[0.01, 0.001, 0.0001]$
  - Activation: [ReLU, Tanh]
- wRNN
  - Gradient Clip Magnitude: $[0, 1.0, 10.0]$
  - Learning Rate $[0.01, 0.001, 0.0001]$

We find the following hyperparameters then resulted in the lowest test MSE for each model:

| Model | Parameter | Sequence Length (T) | | | | |
|---|---|---|---|---|---|---|
| | | 0 | 10 | 30 | 80 | 480 |
| wRNN | Learning Rate | 1e-3 | 1e-3 | 1e-3 | 1e-3 | 1e-4 |
| (n=100, c=6, k=3) | Gradient Magnitude Clip | 1 | 0 | 0 | 1 | 1 |
| | Learning Rate | 1e-3 | 1e-3 | 1e-4 | 1e-4 | 1e-4 |
| iRNN | Gradient Magnitude Clip | 10 | 1 | 1 | 1 | 10 |
| (n=100) | U-initialization | $\mathcal{U}$ | $\mathbf{I}$ | $\mathbf{I}$ | $\mathbf{I}$ | $\mathbf{I}$ |
| | Activation | ReLU | ReLU | ReLU | ReLU | ReLU |
| | Learning Rate | 1e-3 | 1e-4 | 1e-4 | 1e-4 | 1e-4 |
| iRNN | Gradient Magnitude Clip | 1 | 10 | 1 | 1 | 1 |
| (n=625) | U-initialization | $\mathcal{U}$ | $\mathbf{I}$ | $\mathbf{I}$ | $\mathbf{I}$ | $\mathbf{I}$ |
| | Activation | ReLU | ReLU | ReLU | ReLU | ReLU |

Table 5: Best performing hyperparamters for each model on the Copy Task. Gradient clipping 0 means not applied, U initialization $\mathcal{U}$ means Kaming Uniform Initialization.

The total training time for these sweeps was roughly 1,900 GPU hours, with models being trained on individual NVIDIA 1080Ti GPUs. The iRNN models took between 1 to 10 hours to train depending on $T$ and $n$, while the wRNN models took between 2 to 15 hours to train.

**sMNIST.** For the sequential MNIST task we use iRNNs and wRNNs with 256 hidden units. To have a similar number of parameters, we use 16 channels with the wRNN. The models are trained with a batch size of 128 for 120 epochs. The learning rate schedule is defined such that the learning rate is divided by a factor of lr_drop_rate every lr_drop_epoch epochs. We grid search over the following hyperparameters for each model type. We find that the iRNN does not need gradient clipping on this task and achieves smooth loss curves without it. We highlight in yellow the hyperparameters which achieve the maximal performance, and were thus reported in the main text:

- iRNN
    - Learning Rate: [0.001, 0.0001, 0.00001]
    - lr_drop_rate: [3.33, 10.0]
    - lr_drop_epoch: [40, 100]
- wRNN
    - Gradient Clip Magnitude: [0, 1, 10, 100]
    - Learning Rate: [0.001, 0.0001, 0.00001]
    - lr_drop_rate: [3.33, 10.0]
    - lr_drop_epoch: [40, 100]

The total training time for these sweeps was roughly 1000 GPU hours, with models being trained on individual NVIDIA 1080Ti GPUs, iRNN models taking roughly 12 hours each, and wRNN models taking roughly 18 hours each.

**psMNIST.** For the permuted sequential MNIST task, we use the same architecture and training setup as for the sMNIST task. We find that on this task the iRNN requires gradient clipping to perform well and thus include it in the search as follows:

- iRNN
    - Gradient Clip Magnitude: [0, 1, 10, 100, 1000]
    - Learning Rate: [0.001, 0.0001, 0.00001]
    - lr_drop_rate: [3.33, 10.0]
    - lr_drop_epoch: [40, 100]
- wRNN
    - Gradient Clip Magnitude: [0, 1, 10, 100, 1000]
    - Learning Rate: [0.001, 0.0001, 0.00001]
    - lr_drop_rate: [3.33, 10.0]
    - lr_drop_epoch: [40, 100]

In an effort to improve the baseline iRNN model performance, we performed additional hyperparameter searching. Specifically, we tested with larger batch sizes (120, 320, 512), different numbers of hidden units (64, 144, 256, 529, 1024), additional learning rates (1e-6, 5e-6), a larger number of epochs (250), and more complex learning rate schedules (exponential, cosine, one-cycle, and reduction on validation plateau). Ultimately we found the parameters highlighted above to achieve the best performance, with the only improvement coming from training for 250 instead of 120 epochs. Regardless, in Figure 6 we see the iRNN performance is still significantly below the wRNN performance, strengthening the confidence in our result. The total compute time for these sweeps was roughly 1,900 GPU hours with models being trained on individual NVIDIA 1080 Ti GPUs. The iRNN models took rougly 12 hours each, with wRNN models taking roughly 18 hours each.

For each of the wRNN models in Figure 7, the same hyperparameters are used as highlighted above and found to perform well. The wRNN is tested over combinations of $n = (16, 36, 64, 144) \times c = (1, 4, 16, 32)$, and the best performing models for each parameter count range are displayed. For the iRNN, we sweep over larger batch sizes (128, 512), learning rates (1e-3, 1e-4, 1e-5) and gradient clipping magnitudes (0, 1, 10, 100) in order to stabalize training, displaying the best models.

**nsCIFAR10.** For the noisy sequential CIFAR10 task, models were trained with a batch size of 256 for 250 epochs with the Adam optimizer, $\mathrm{lr\_drop\_epoch} = 100$, and $\mathrm{lr\_drop\_rate} = 10$. We found gradient clipping was not necessary for the wRNN model on this task, and thus perform a grid search as follows, with the best performing settings highlighted:

- iRNN
    - Learning Rate: [0.001, 0.0001, 0.00001]
    - Number hidden units (n): [144, 256, 529, 1024]

- wRNN
    - Learning Rate: $[0.001, \boxed{0.0001}, 0.00001]$
    - Number hidden units (n): $[144, \boxed{256}]$

The total compute time for these sweeps was roughly 1,600 GPU hours with models being trained on individual NVIDIA 1080 Ti GPUs. The iRNN models took roughly 15 hours each, with wRNN models taking roughly 22 hours each.

**Adding Task.**  For the adding task we again train the model with batch sizes of 128 for 60,000 batches using the Adam optimizer. For this task models are trained with a mean squared error loss. For both the iRNN and wRNN, we then grid-search over the following hyperparameters:

- Gradient Clip Magnitude: $[0, 1, 10, 100, 1000]$
- Learning Rate: $[0.01, 0.001, 0.0001]$

In Table 6 we report the best performing parameters for each task setting. The total training time for these sweeps was roughly 1,200 GPU hours. The iRNN models took between 1 to 7 hours, while the wRNN models took 6 to 12 hours each .

| Model | Parameter | Sequence Length (T) | | | | |
|---|---|---|---|---|---|---|
| | | 100 | 200 | 400 | 700 | 1000 |
| wRNN | Learning Rate | 1e-3 | 1e-4 | 1e-4 | 1e-4 | 1e-4 |
| (n=100, c=27, k=3) | Gradient Magnitude Clip | 100 | 100 | 1 | 100 | 10 |
| iRNN | Learning Rate | 1e-3 | 1e-3 | 1e-3 | 1e-4 | 1e-3 |
| (n=100) | Gradient Magnitude Clip | 1000 | 100 | 10 | 100 | 1 |

Table 6: Best performing settings on the Adding Task. Gradient clipping 0 means not applied. We note that the iRNN failed to solve the task meaningfully for lengths T = 700 & 1000, thus the hyperparameters found here are not significantly better than any other combination for those settings.

**Ablation.**  For the ablation results in Table 4, we first report the test MSE of the best performing wRNN and iRNN models from the original Copy Task grid search (identical to those in Figure 3). We then added the ablation settings to the grid search, and trained the models identically to those reported above in the **Copy Task** section. This resulted in the final complete grid search:

- iRNN
    - Gradient Clip Magnitude: $[0, 1.0, 10.0]$
    - $\mathbf{U}$ Initialization: $[\mathbf{I}, \mathcal{U}(-\frac{1}{\sqrt{n}}, \frac{1}{\sqrt{n}}), \Sigma]$
    - $\mathbf{V}$ Initialization: $[\mathcal{N}(0, 0.001), \text{Sparse-Identity}]$
    - Learning Rate: $[0.01, 0.001, 0.0001]$
    - Activation: [ReLU, Tanh]
- wRNN
    - Gradient Clip Magnitude: $[0, 1.0, 10.0]$
    - $\mathbf{u}$ Initialization: $[\text{Dirac}, \mathcal{U}(-\frac{1}{\sqrt{n}}, \frac{1}{\sqrt{n}}), \mathbf{u}\text{-shift}]$
    - $\mathbf{V}$ Initialization: $[\mathcal{N}(0, 0.001), \text{Sparse-Identity}]$
    - Learning Rate $[0.01, 0.001, 0.0001]$

where Sparse-Identity refers to the $\mathbf{V}$ initialization described in section 2. In the case of the iRNN, the $\mathbf{V}$ matrix is defined to have 1 channel for the purpose of this initialization. The Dirac initialization is equivalent to an identity initialization for a convolutional layer and is implemented using the Pytorch function: `torch.nn.init.dirac_`. We report the best performing models from this search in Table 4.

For the wRNN (-$\mathbf{u}$-shift-init), we note that for the sequence lengths $T$ where the wRNN model does solve the task (MSE $\leq 1 \times 10^{-10}$), i.e. T=0 & 10, the best performing models always use the $\mathbf{u}$

initialization $\mathcal{N}(0, 0.001)$ and appear to learn to exhibit traveling waves in their hidden state, while the dirac initialization always performs worse and does not exhibit waves.

## C  ADDITIONAL RESULTS

**Performance Means & Standard Deviations.** In the main text, in order to make a fair comparison with prior work, we follow standard practice and present the test performance of each model with the corresponding best validation performance. In addition however, we find it beneficial to report the distributional properties of the model performance after multiple random initalizations. In Table 7 we include the means and standard deviations of performance from 3 reruns of each of the models.

| Task | Metric | iRNN | wRNN |
|---|---|---|---|
| Adding T=100 | MSE
Solved iter. | $1.40\times10^{-5} \pm 4.21\times10^{-6}$
$11,500.00 \pm 2,910.33$ | $6.12\times10^{-5} \pm 7.95\times10^{-5}$
$233.33 \pm 57.74$ |
| Adding T=200 | MSE
Solved iter. | $5.13\times10^{-5} \pm 1.66\times10^{-5}$
$21,000.00 \pm 4,582.58$ | $8.54\times10^{-5} \pm 7.91\times10^{-5}$
$1,000.00 \pm 0.00$ |
| Adding T=400 | MSE
Solved iter. | $7.70\times10^{-2} \pm 8.49\times10^{-2}$
$30,000.00 \pm -$ | $1.59\times10^{-4} \pm 1.07\times10^{-4}$
$1,333.33 \pm 577.35$ |
| Adding T=700 | MSE
Solved iter. | $0.163 \pm 2.08\times10^{-3}$
$\times$ | $5.29\times10^{-5} \pm 3.19\times10^{-5}$
$3,000.00 \pm 0.00$ |
| Adding T=1000 | MSE
Solved iter. | $0.160 \pm -$
$\times$ | $4.36\times10^{-5} \pm 1.91\times10^{-5}$
$1,666.67 \pm 577.35$ |
| sMNIST | Test Acc. | $98.20 \pm 0.32$ | $97.30 \pm 0.34$ |
| psMNIST | Test Acc. | $90.85 \pm 1.47$ | $96.60 \pm 0.10$ |
| nsCIFAR10 | Test Acc. | $51.80 \pm 0.54$ | $54.70 \pm 0.42$ |

Table 7: Mean and standard deviation of model performance over 3 random initializations for the best performing models in each category. We see model performance is consistent with the best performing models reported in the main text. The $\times$ means the models never solved the task after 60,000 iterations, and ($\pm$ -) means that the other 2/3 random initalizations also did not solve the task after 60,000 iterations or crashed.

**Visualizing Multiple Channels.** To visualize how activations might flow between channels, we include a visualization of 15 of the channels for the wRNN on the sMNIST task in Figure 9.

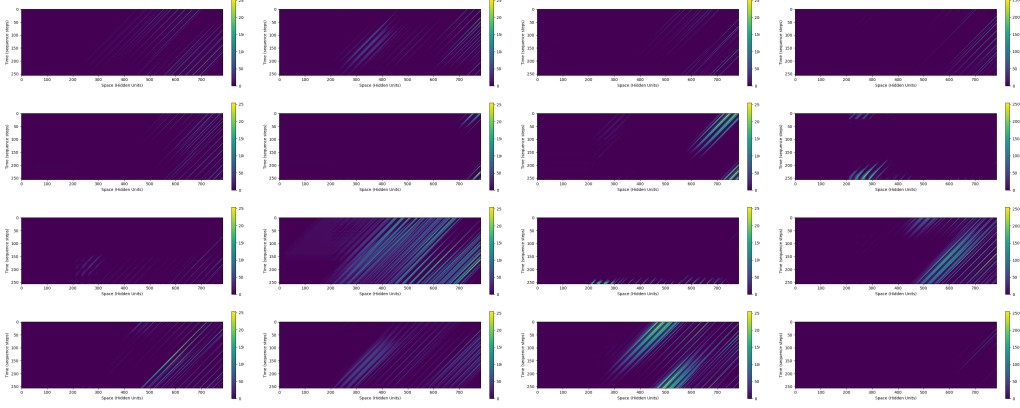

Figure 9: Visualization of 15 separate hidden state channels for a wRNN trained on the sMNIST task. We see wave activity differs between channels and has the potential to flow between channels although it is difficult to precisely quantify due to their lack of predefined ordering.

**Waves of Multiple Velocities.** To compliment our visualization of waves in the main text, we show in Figure 10 multiple distinct channels of a single wRNN trained on sMNIST. For such a network, we randomly initlize the velocity $\nu$ of each channel.

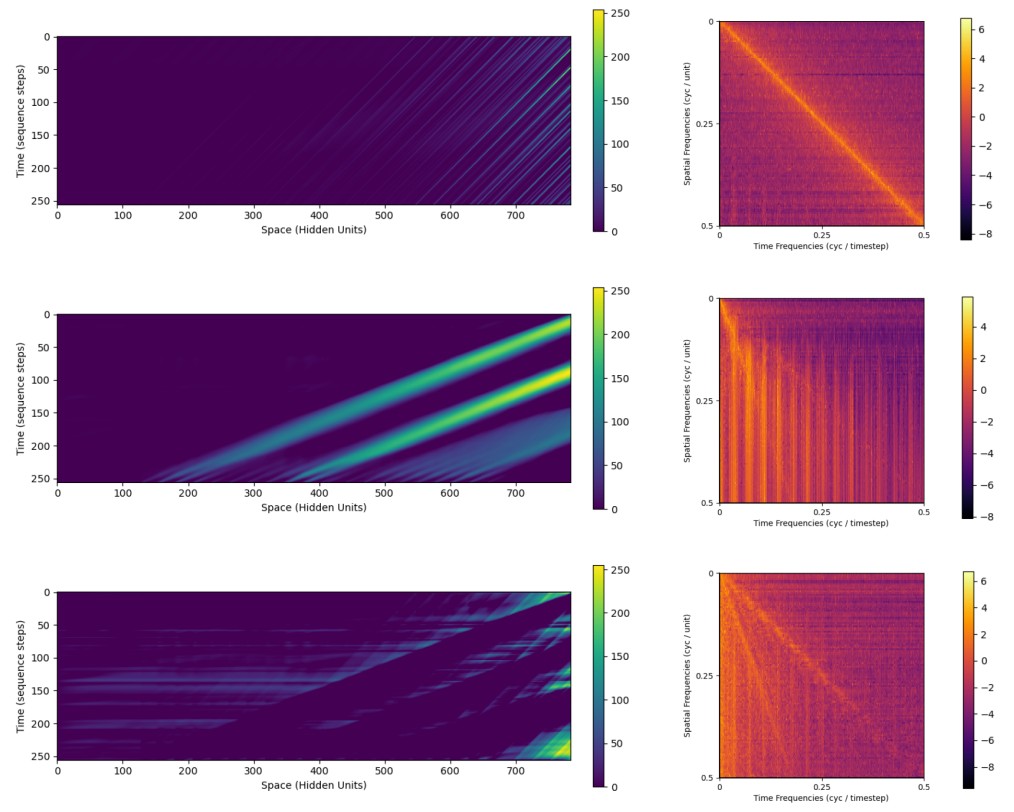

Figure 10: Depiction of traveling waves of multiple speeds within the same model. Each row shows a separate channel of the hidden state for a single model. The speed of the waves are given by the slopes of the lines of high activation.

**Unsorted Visualization of iRNN Hidden State.** To ensure that our sorting procedure of iRNN neurons in Figure 2 did not artificially destroy localized activity which may have otherwise been visible, in the following Figure 11 we plot the same set of neural activations, with neurons unsorted for the iRNN. We see that there is less structure as expected, and still no wave-like activity.

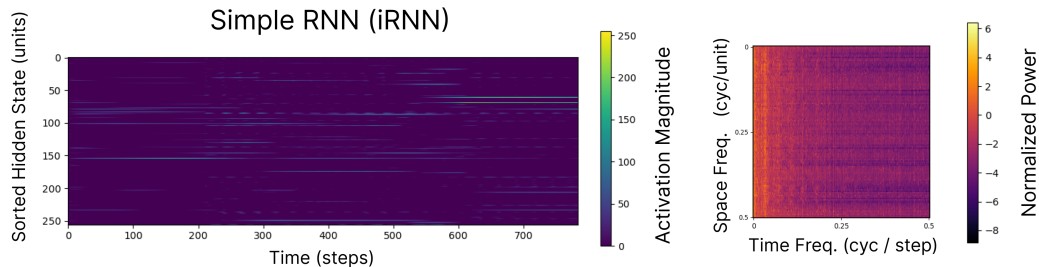

Figure 11: Visualization of hidden state (left) and associated 2D Fourier transform (right) for an iRNN after training on the sMNIST task. This is identical to Figure 2 without the sorting of iRNN neurons by onset time.

**Fourier Analysis Calibration**   To demonstrate the relationship between the speed of traveling waves and the resulting slope of peak power in the 2D Fourier transform, in Figure 12 we compute the 2D Fourier transform of synthetic data where we have induced known velocities. As expected, we see the slope matches the observed wave velocities. We note that we only plot the magnitude of positive frequency components in this work since all signals are real values and thus the remaining components are simply the complex conjugates of these components.

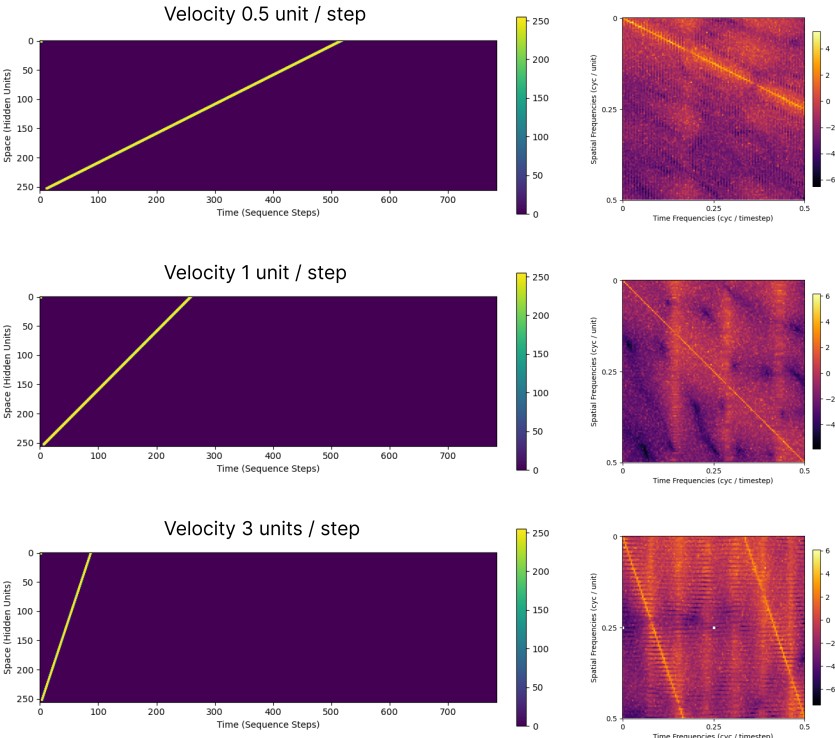

Figure 12: Examples of synthetic waves with known velocities (left) and the resulting 2D Fourier Transforms (right). We see indeed the slope of the peak magnitude in Fourier space corresponds to the wave velocities.

**Additional Ablation: Locally Connected RNN.**   In order to test if the emergence and maintenance of waves requires the weight sharing of the convolution operation, or is simply due to local connectivity, we perform an additional experiment where we use the exact same model as the default wRNN, however we remove weight sharing across locations of the hidden state. This amounts to replacing the convolutional layer with an untied 'locally connected' layer. In practice, when initialized with same the shift-initialization we find that such a model does indeed exhibit traveling waves in its hidden state as depicted in Figure 13. Although we notice that the locally connected network does train to comparable accuracy with the convolutional wRNN on sMNIST, we present this preliminary result as a simple ablation study and leave further tuning of the performance of this model on sequence tasks for future work.

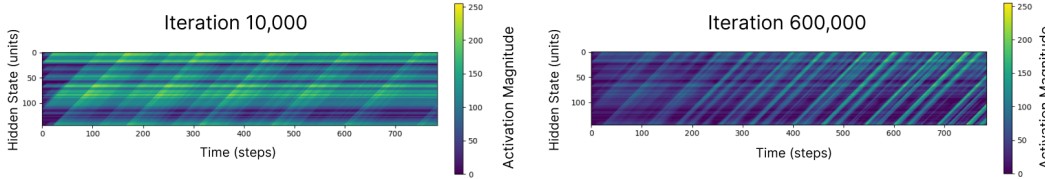

Figure 13: Hidden state of a locally connected RNN showing the existence of traveling waves. These results imply that it is simply local connectivity which is important for the emergence of traveling waves rather than shared weights.

**Additional Baseline: Linear Layer on MNIST.** One intuitive explanation for the performance of the wRNN is that the hidden state 'wave-field' acts like a register or 'tape' where the input is essentially copied by the encoder, and then subsequently processed simultaneously by the decoder at the end of the sequence. To investigate how similar the wRNN is to such a solution, we experiment with training a single linear layer on flattened MNIST images, equivalent to what the decoder of the wRNN would process if this 'tape' hypothesis were correct. In Figure 14 we plot the results of this experiment (again showing the best model from a grid search over learning rates and learning rate schedules), and we see that the fully connected layer achieves a maximum performance of $92.4\%$ accuracy compared with the $97.6\%$ accuracy of the wRNN model.

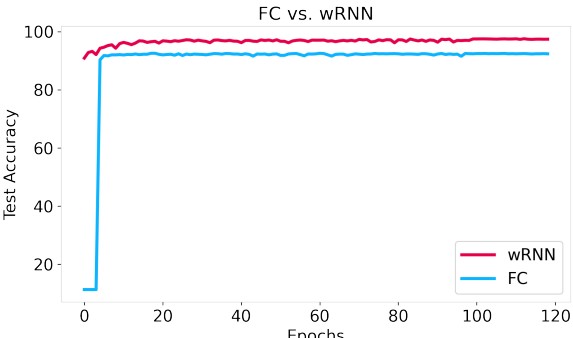

Figure 14: Training curves for a fully connected layer trained on flattened MNIST images (FC) versus the standard wRNN presented in the main text. We see the wRNN still performs significantly better, however the FC model does mimic the rapid learning capability of the wRNN, suggesting that the wRNN's learning speed may be partially attributable to the wave-field's memory-tape-like quality.

**Additional Ablation: Frozen Encoder & Recurrent Weights.** To further investigate the difference between the wRNN model and a model which simply copies input to a register, we propose to study the relative importance of the encoder weights $\mathbf{V}$ and recurrent connections $\mathbf{U}$ for the wRNN. We hypothesized that the wRNN may preserve greater information in its hidden state by default, and thus may not need to learn a flexible encoding, or perform recurrent processing. To test this, we froze the encoder and recurrent connections, leaving only the decoder (from hidden state to class label) to be trained. In Figure 15 we plot the training curves for a wRNN and iRNN with frozen $\mathbf{U}$ and $\mathbf{V}$. We see that the wRNN performs remarkably better then the iRNN in this setting (89.0% vs. 44.3%), indicating that the wave dynamics do indeed preserve input information by default far better than standard (identity) recurrent connections.

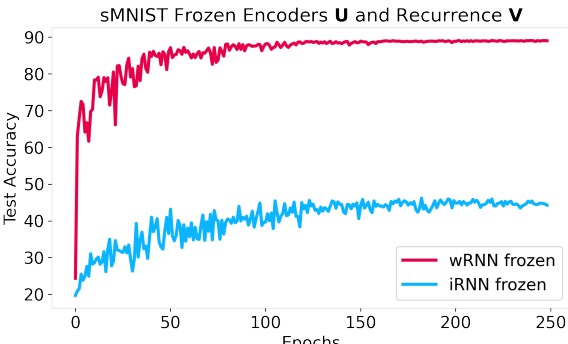

Figure 15: Training curves for wRNN and iRNN models with frozen encoder and recurrent connections ($\mathbf{U}$ & $\mathbf{V}$) on the sMNIST task. We see that the wRNN performs drastically better, indicating that the wRNN requires significantly less flexibility in its encoder and recurrent connections in order to achieve high accuracy.

**Additional Visualizations of Copy Task.** Here we include additional visualizations of the smaller iRNN (n=100) for the copy task. We see that it performs significantly worse than the larger iRNN (n=625) displayed in the main text, unable to recall sequences for any length of time.

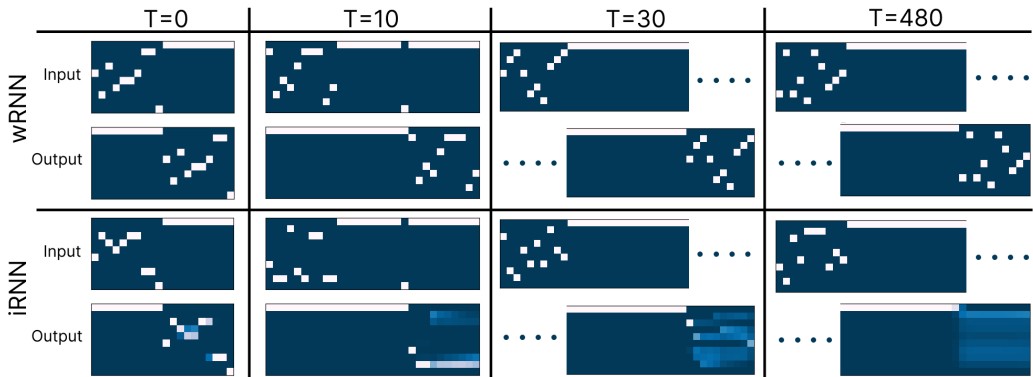

Figure 16: Examples from the copy task for wRNN (n=100, c=6) and iRNN (n=100). We see the iRNN loses significant accuracy after T=10. These results clearly show that the iRNN does not have the appropriate machinery for storing memories over long sequences while the wRNN does.

**On the Emergence of Traveling Waves.** In this section we expand on the emergence of traveling waves in recurrent neural networks with different connectivity and initialization schemes. Specifically, in Figure 18 we show the hidden state visualization for models with varying initalizations. We see that models with shift-initialization exhibit waves directly from initialization, while randomly initialized convolutional models do not initially exhibit waves but learn to exhibit them during training, and identity initialized models never learn to exhibit waves. Furthermore, in Figure 17 we show the respective training curves for randomly initialized and identity initialized wRNNs. We see that the randomly initialized wRNN achieves higher final accuracy in correspondence with the emergence of traveling waves, reinforcing the conclusion that traveling waves are ultimately beneficial for sequence modeling performance.

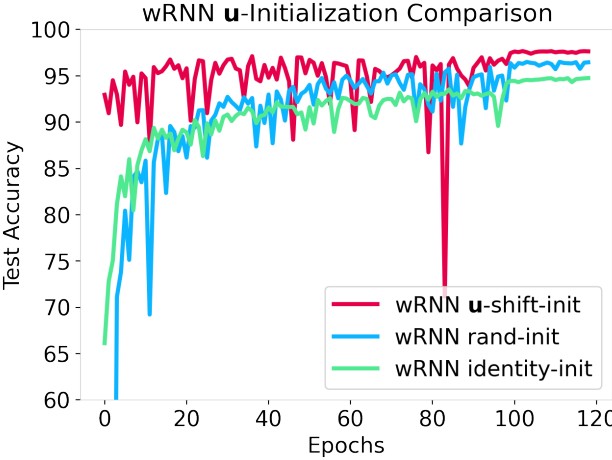

Figure 17: Training curve on the sMNIST task for a wRNN with three different initialization schemes: $\mathbf{u}$-shift (default), random ($\mathcal{U}(-\frac{1}{\sqrt{n}}, \frac{1}{\sqrt{n}})$), and identity (dirac). We see that the random initalization does not have the same rapid learning speed as the $\mathbf{u}$-shift initialization, however, it does still achieve significantly higher final accuracy than the identity initialization, implying traveling waves are beneficial to performance.

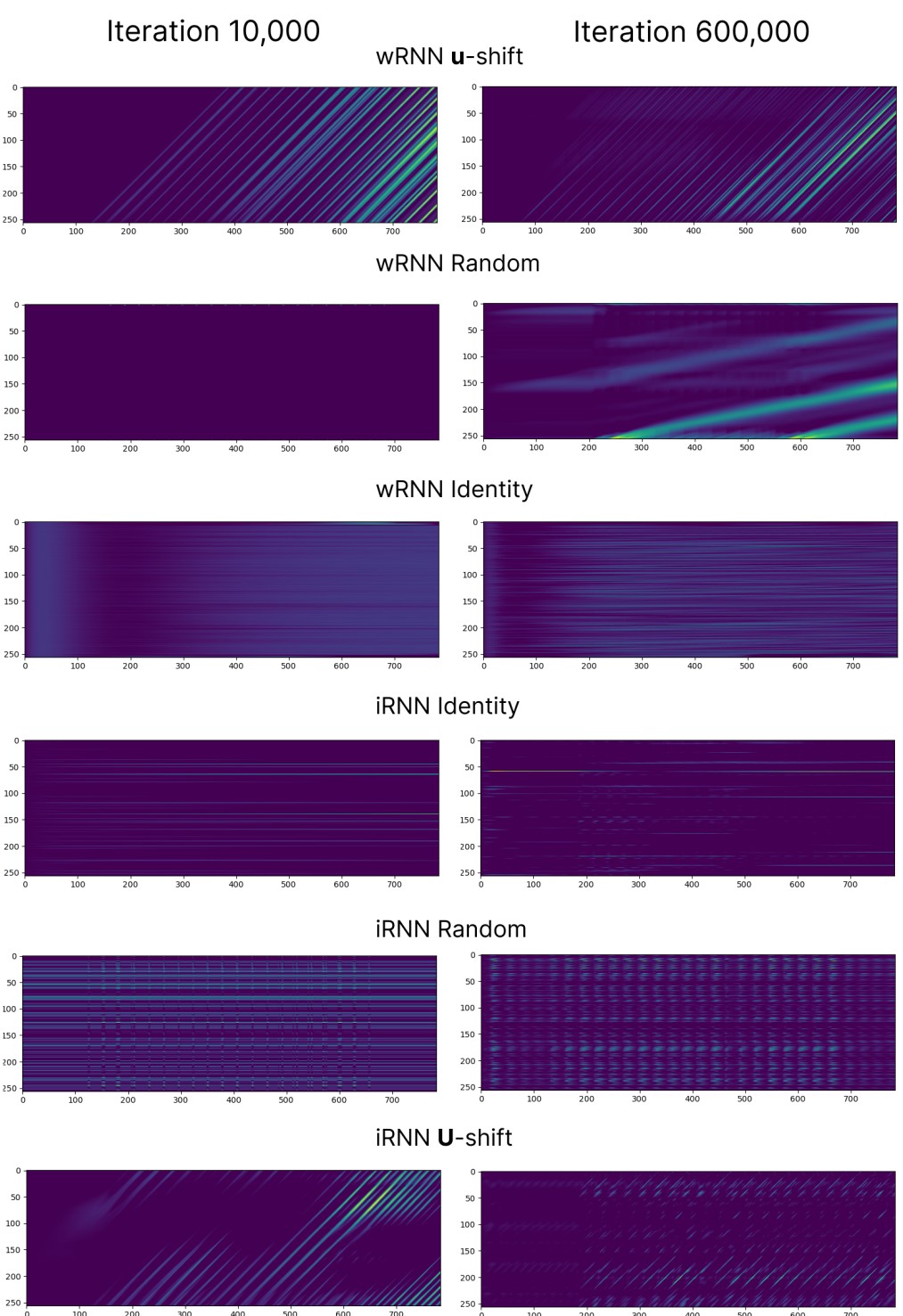

Figure 18: Visualization of hidden state (y-axis is hidden units) over timesteps (x-axis) for a variety of different models and initalizations for **U**. We see that the wRNN with **u**-shift initialization achieves the most consistent waves throughout training. Interestingly, some other models learn to achieve traveling waves despite not having them at initalization (wRNN with random (kaming uniform) initialization); while other models, (iRNN with **U**-shift) initially have stronger traveling waves, and slowly lose them throughout training. We see the wRNN with identity (dirac) initialization never learns waves despite using convolutional recurrent connections.

**Waves with Variable Velocity.** As described in the related work, there is a relation between the ideas presented in this paper and the notion of 'time cells' from the neuroscience literature. One interpretation of time cells is that they could be represented by a wave which slows down as it progresses over the cortical surface. To explore the impact of such a modification, we implemented a locally-connected wRNN architecture which is initialized such that the velocity $\nu$ systematically decreases for each $i$'th row of the $\Sigma$ matrix as $\frac{1}{i}$. In such a model the wave speed could indeed be interpreted to decrease as it progresses. In Figure 19 we plot the hidden state of such a model after being trained on the sMNIST task. We see that waves indeed travel significantly more slowly, and appear to have curved trajectories, implying 'slowing'. In terms of performance, we see that the model reaches roughly the same test accuracy as the default wRNN presented in the main text (%97.76), suggesting that such a modification could be a performant alternative to study in the future.

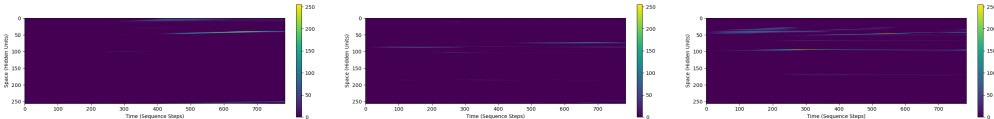

Figure 19: Visualization of the hidden state of a locally connected wRNN model, trained on sMNIST, initialized with velocities that decrease as a function of position, emulating a wave which slows down as it propagates. We see that the waves do indeed appear to have velocities which change as a function of position in the hidden state, and the model still learns to perform well on the task.

**Impact of Hidden State Size.** Due to cyclic boundary conditions, the wRNN should in theory be able to handle arbitrary time delays with constant computational complexity (i.e. waves simply propagate in circles until decoded). Wave-RNNs should therefore be able solve sequence tasks with a number of steps significantly greater than the number of hidden units. For example, we have found that models with as little as 49 hidden units (6 channels) can solve the copy task with a delay of 1000 time steps. Despite this theory, in practice, we have found models with larger hidden states still typically perform better on longer sequence tasks. We demonstrate this in Table 8 below. We believe this divergence from theory to be primarily due to optimization challenges. In our experiments we found runtime to correlate significantly sub-linearly with hidden state size, with the majority of computational complexity arising from iterating over sequence length.

| Seq. Length (T) | 50 | 100 | 200 | 500 |
|---|---|---|---|---|
| n_hid = 9 | 8e-3 | 1e-2 | 1e-2 | 2e-4 |
| n_hid = 16 | 1e-3 | 2e-2 | 6e-3 | 4e-4 |
| n_hid = 25 | 1e-4 | 2e-3 | 2e-3 | 1e-5 |
| n_hid = 49 | 1e-8 | 8e-5 | 9e-6 | 2e-4 |
| n_hid = 100 | **3e-9** | **7e-7** | **4e-7** | **1e-6** |

Table 8: Test loss on the copy task for the wRNN for different Hidden State sizes (n_hid) vs. Sequence Lengths. We see the larger hidden state sizes almost consistently achieve lower error.

