# OpenReview forum: "Traveling Waves Encode The Recent Past and Enhance Sequence Learning"
_ICLR.cc/2024/Conference — ICLR 2024 poster_

### Official Review · Reviewer_6SWP · 2023-10-30

**Soundness:** 3 good
**Presentation:** 3 good
**Contribution:** 4 excellent
**Rating:** 6
**Confidence:** 3

**Summary:**

Over the past decade, recurrent neural networks, trained by backpropagation through time, have been used to infer mechanisms employed by networks of biological neurons to perform cognitive tasks. Yet, most existing literature on biological RNNs focus on attaining fixed or slow points at steady-state, and neglects the other well-known solution -- oscillations. On the other hand, traveling waves have been observed in the brain and studied for multiple decades. Waves in general are well-understood mechanistically, but their computational role in the brain remains a hot debate. Here, the authors bridge this very obvious gap by introducing an RNN architecture that produces traveling waves. Wave-RNNs store waves in multiple rings (which they call "channels") and seemingly perform reasonably well across a variety of sequence-based tasks.

I am supportive of the general idea and the rigor of this work. However, the model introduction was very hard to parse and requires several rereads and returning to after reading future sections in order to reconcile any confusion.

**Strengths:**

This work is timely and important for the biologically-plausible RNN community to acknowledge the possibility of oscillatory/wave-like solutions that can arise from gradient-based training. The work is rigorous and well-motivated.

**Weaknesses:**

While the general narrative makes sense, I find the introductory narrative very hard to understand.

(1) Since $\mathbf{u}$ has dimensions $c \times c \times f$, this means that there are convolutional kernels between rings, but the intuition provided does not seem to acknowledge this.

(2) The usage of identity-initialized RNNs was also not clear. From the parameter count of such models, it seems like every element of the weight matrix is being optimized. That is not mentioned anywhere in the text (or if it is, it should be highlighted more clearly). Indeed, that is the case for Le et al 2015, but this would detract from the bump model which the authors originially intended to be the main baseline model to compare with wave-RNNs.

(3) The way wave-RNNs are named is confusing and inefficient. At some times, it is labeled as $n = 100, c=6$ to represent $n$ as the total number of neurons and $c$ as the number of channels. This means that there are 16 neurons per channel, and where does the last 4 neurons go (how does the floor function in page 3 work)? At other times it is called $16c$ which I assume refers to the same thing. In both ways of naming, there is no mention about the dimensionality of the kernel -- I suspect it may not be constant because of the remaining 4 neurons, but there has to be a better way to label everything.

(4) In Figure 2, neurons in the iRNN are sorted according by time of maximum activation. How are the neurons in the wRNN sorted and where is the channel separation?

(5) The training curves in Figures 3,5,6 are extremely transient and subject to random "resets" in performance. This means that in a single training session, the efficiency of training depends (by luck) on number of resets that happen, which is very apparent in Figure 3. This makes any conclusions drawn about training efficiency unconvincing. More models should be trained and the loss curves averaged.

**Questions:**

(1) Did the authors really initialize $\mathbf{V}$ to be zero (as claimed on page 3)? I think it is a typo and they actually meant initializing $\mathbf{b}$ to be zero instead?

(2) There seems to be no additional point in training wRNN + MLP? It provides a small improvement and a single sentence explaining that linear decoding is a bottleneck (which is in fact an important bottleneck to prevent overfitting in neural data) -- am I missing something?

---

> ### Author Response · Authors · 2023-11-23
> **Response to Reviewer 6SWP (1/2)**
>
> ## Summary
> We thank the reviewer for their acknowledgment of the biological relevance of our work and for the contribution of a model which exhibits wave-like solutions from gradient-based training.
>
> Below we address the reviewer’s proposed weaknesses and questions.
> > (1) Since $\mathbf{u}$ has dimensions $c \times c \times f$, this means that there are convolutional kernels between rings, but the intuition provided does not seem to acknowledge this.
>
> Indeed the convolutional kernels are fully parameterized and therefore support waves between channels. We believe the intuition provided in our introduction still supports this. Specifically, since the mapping between channels and the cortical surface is not firmly fixed, it would fit within our model that channels may be laid out parallel on the cortical surface, and such cross-channel communication could then again simply be implemented by local lateral connectivity. We appreciate the reviewer bringing this up, and will strive to make the multi-channel nature of the convolution more clear.
>
> > (2) The usage of identity-initialized RNNs was also not clear. From the parameter count of such models, it seems like every element of the weight matrix is being optimized. That is not mentioned anywhere in the text (or if it is, it should be highlighted more clearly). Indeed, that is the case for Le et al 2015, but this would detract from the bump model which the authors originally intended to be the main baseline model to compare with wave-RNNs.
>
> In both settings all parameters are being optimized as this is necessary for both models to achieve optimal performance. We appreciate the reviewer highlighting that this is not clear, and we have modified the text to reflect this. As we found and demonstrated empirically in Figure 2, despite this full parameterization and optimization, the iRNN still learns a solution which resembles a static ‘bump’ solution, and therefore we believe it to still fit within our motivation.
>
> > (3) The way wave-RNNs are named is confusing and inefficient. At some times, it is labeled as $n = 100, c=6$ to represent  as the total number of neurons and $c$ as the number of channels. This means that there are 16 neurons per channel, and where does the last 4 neurons go (how does the floor function in page 3 work)? At other times it is called $16c$ which I assume refers to the same thing. In both ways of naming, there is no mention about the dimensionality of the kernel -- I suspect it may not be constant because of the remaining 4 neurons, but there has to be a better way to label everything.
>
> We thank the reviewer for highlighting potential confusion with respect to the total number of neurons and our notation. In our plots, $n$ refers to the number of neurons per channel, so the total number of neurons is $n \times c$. We have modified the text to make this more clear.
>
> > (4) In Figure 2, neurons in the iRNN are sorted according by time of maximum activation. How are the neurons in the wRNN sorted and where is the channel separation?
>
> There is no sorting of the wRNN neurons, they inherently have spatial locality due to the convolution operation. We therefore simply plot the hidden state vector in the order of the array over which the convolution is applied.
>
> > (5) The training curves in Figures 3,5,6 are extremely transient and subject to random "resets" in performance. This means that in a single training session, the efficiency of training depends (by luck) on number of resets that happen, which is very apparent in Figure 3. This makes any conclusions drawn about training efficiency unconvincing. More models should be trained and the loss curves averaged.
>
> While we agree that the dramatic increases in training loss are less than ideal, we would like to make two notes which may at least temporarily ease the reviewers concerns. First, we note that the y-axis is on a logarithmic scale, therefore although it may look like the loss is 'resetting’, in practice these changes in performance are not as significant as they may first appear. Secondly, we note that all of the believably harmful 'resets’ which happen occur for the wRNN model, which still learns orders of magnitude faster than the iRNN.
>
> Although we unfortunately do not have time to rerun all of the loss curves and average them for this rebuttal we would certainly be happy to do so for the camera ready if accepted, and we can assure the reviewer that for the reruns we did during development, we saw the same results consistently — the wRNN trained orders of magnitude faster than the iRNN.

---

> > ### Author Response · Authors · 2023-11-23
> > **Response to Reviewer 6SWP (2/2)**
> >
> > > Questions: (1) Did the authors really initialize $\mathbf{V}$ to be zero (as claimed on page 3)? I think it is a typo and they actually meant initializing $\mathbf{b}$ to be zero instead?
> >
> > We thank the reviewer for highlighting their confusion. As noted on page 3, we initalize V to be zero *except for a single row which is the identiy*. This means that it the input is effectively driving each single channel independantly from a single source location. In Appendix B The reviewer can see the exact code which is used to initalize the matrices.
> >
> > > (2) There seems to be no additional point in training wRNN + MLP? It provides a small improvement and a single sentence explaining that linear decoding is a bottleneck (which is in fact an important bottleneck to prevent overfitting in neural data) -- am I missing something?
> >
> > We believe the linear decoding bottleneck is indeed highly important to emphasize since this is a major distinction between the wRNN and prior work with which our model is compared (e.g. the best performing models in Tables 2 & 3 are highly non-linear). Specifically, due to the linear encoders in our model, there is effectively very little depth to our model, making comparison with the very deep counterparts listed less fair. To partially account for this we have added the MLP experiment, but ideally would add a multi-layer wRNN experiment in the future. This serves as an intermediate step to show that indeed the linear decoder is a bottleneck. In practice we do not find the model to be overfitting in any significant way yet and therefore politely disagree with the reviewers assertion that a linear decoder is necessary in this setting.

---

### Official Review · Reviewer_2PWQ · 2023-10-31

**Soundness:** 3 good
**Presentation:** 2 fair
**Contribution:** 2 fair
**Rating:** 6
**Confidence:** 4

**Summary:**

The present study uses a neurally inspired convolutional recurrent network model with traveling wave states to accomplish a set of tasks. The authors claim the traveling wave recurrent network outperforms the one without waves.

**Strengths:**

The present study is a good and novel example of brain-inspired computation in that it uses the wave RNN (inspired by the brain) to implement a set of benchmark tasks. It also considers a set of experiments that support the wave RNN indeed outperforms its counterparts without traveling waves. It provides us insight that structured spatiotemporal dynamics can have its advantages in real applications.

**Weaknesses:**

### Major
- Although the concept of this study is novel, I feel the present paper can be strengthened by conducting a deeper analysis to show why internally generated traveling waves in RNNs can improve those computational tasks. For example, since the network model is small (I see the limitation discussed in the end), the authors could perform a dimensionality analysis to demonstrate the network's evolution in the low-dimensional manifold. I do see the author providing an intuitive explanation of the benefit of traveling waves in the introduction, and showing neurons' response in Fig. 2, but they are not enough from my point of view.

- The illustrative example in Fig. 1 (middle) is probably not a perfectly matching example in explaining the benefit of traveling waves. The Fig. 1 illustrates an example of a two-way wave equation, however, the traveling wave in the RNN is only a one-way wave equation that the network state moves in a single direction. I am not clear about why the traveling wave could maintain the spatiotemporal information.

I'd like to raise my rating if the two concerns can be properly addressed.

### Minor
- Eq. 2: it seems that the matrix $\Sigma$ misses the time step $\Delta t$ in discretizing Eq. 1. Otherwise the speed $v$ cannot exceed 1 (the term $1-v$ in $\Sigma$). I mean you can absorb the $Delta t$ into a new parameter in Eq. 2 without explicitly expressing it, but in this case the $v$ in Eq. 2 is not the same $v$ in Eq. 1.

- Below Eq. 2: I am confused about what the input channel means.

- Below Eq. 3: it is not clear how the convolution between $u$ with dimension [c,c,f]  and $h$ with dimension [c,n'] is calculated.

**Questions:**

- Do you need to retrain the RNN to copy sequences with different lengths, or produce waves with different speeds?

---

> ### Author Response · Authors · 2023-11-23
> **Response to Reviewer 2PWQ (1/2)**
>
> ### Summary
> We thank Reviewer 2PWQ for their time spent with our paper, for acknowledgement of the novelty of our submission, and for their helpful comments which have helped make the presentation of our work more clear. In the following we explain how we have attempted to address their major (and minor) concerns in our updated submission.
>
> ### Major Point 1: Deeper Understanding
> We appreciate the reviewer noting how the clarity of our presentation could be improved. In response to this, we have updated the intuition provided in our introduction to give a better understanding of the working mechanism of our model which we believe may partially address the reviewer's concern regarding a lack of analysis. Specifically, as pointed our by reviewer h1zv, it is potentially easiest to understand the success of the wRNN model as a having a hidden state in the form of a 'register' or 'stack' where information is sequentially written or pushed. The propagation of waves then serves to prevent overwriting of past input, allowing for superior invertible memory storage.
>
> Pursuant to this, we greatly appreciate the reviewer's suggestion to perform low dimensional analysis of the hidden state dynamics, however we believe that with an intuitive understanding of the dynamics as described above this may not be necessary. Specifically, since we can interpret the hidden state as a set of different `stacks' (one for each channel) where information largely flows within each individual channel separately, the low-dimensional manifold over which the hidden state will evolve is explicitly biased towards these 1-D circular topologies that we have plotted in Figure 2, and in the appendix. We encourage the reviewer to look at Figure 9 of the updated appendix which includes visualization of all channels, as this reinforces the understanding that in general inter-channel communication is relatively minor, and the subspaces of interest for hidden state evolution are exaclty those plotted, as this is the intended design of the network.
>
> ### Major Point 2: Figure 1
> We thank the reviewer for highlighting the potential confusion caused by this figure. Indeed in response to these concerns and those of Reviewer h1zv, we have updated Figure 1 to better match the 1-dimensional structure of the wRNN hidden state. Additionally, we believe this new figure better reflects the new intuition provided in the introduction, suggesting to interpret the hidden state as a 'register' or 'stack' where information is written/pushed. We believe this may help to better convey the understanding of the model that the reviewer felt was lacking and thereby additionally contribute to addressing the reviewer's first major point.
>
> ### Minor Point 1: Missing $\Delta t$
> We thank the reviewer for highlighting the skipped steps in our derivation of the wRNN. Indeed we had assumed that the timestep was absorbed into $\nu$ in equation 2. The full derivation is as follows:
> $$ h_{t+1}(x)  = h_t(x) + \Delta t \left[\nu \frac{\partial h(x, t)}{\partial x}\right] $$
> $$  h_{t+1}(x) \approx h_t(x) + \Delta t \nu \left[h_t(x+1) - h_t(x)\right] $$
> $$  h_{t+1}(x) = (1 - \Delta t \nu) h_t(x) + \Delta t \nu  h_t(x+1) $$
> In the updated submission we have changed the velocity in equation 2 to $\nu'$ and noted that this incorporates the timestep.
>
> ### Minor Point 2: Input Channels
> We use the term 'input channels' in the standard sense employed when describing convolutional neural network layers. The only difference is that in our setting the convolution operates on an RNN hidden state vector, rather than an input image or feature map. By channels of the hidden state, we then refer to subsets of the hidden state obtained by reshaping the traditional $n$ dimensions into $c$ separate $n' = \lfloor\frac{n}{c}\rfloor$ dimensional subsets (as described after Equation 3). In traditional convolutional neural networks, these subsets of the input signal are typically called the input channels, and the resulting subsets of the output vector (feature maps) are often called the output channels. This naming extends to the convolutional kernel as well, which has weights for each input-output pair of channels respectively. This then extends into the reviewer's minor point 3 below.
>
> ### Minor Point 3:
> The convolution between the kernel $\mathbf{u}$ with dimensionality [$c_{out} \times c_{in} \times f$] and the input  with dimensionality [$c_{in} \times n'$], can be understood as computing $c_{out}$ separate convolutions of shape [$c_{in} \times f$] with signal  [$c_{in} \times n'$], where each convolution has a different set of weights -- this is identical to convolution in a standard convolutional neural network layer. In our setting, since we are employing this in a recurrent neural network, the input and output dimensionality must be the same and thus $c_{out} = c_{in} = c$.

---

> > ### Author Response · Authors · 2023-11-23
> > **Response to Reviewer 2PWQ (2/2)**
> >
> > ### Question: Retraining
> > > Do you need to retrain the RNN to copy sequences with different lengths, or produce waves with different speeds?
> >
> > Since the speed of each wave is determined by the parameters of the convolution kernel (which are optimized through training), in order to get waves with different speeds, the network will need to be retrained. In future work, one could imagine alternative 'fast-weight' methods for determining the convolutional kernel parameters (such as with a hypernetwork) where retraining would not be necessary and the speed of waves would be a function of the input itself. This would certainly be a very interesting extension to our model. With respect to generalization to different sequence lengths, we have not performed these experiments and therefore cannot comment with certainty, however we believe there could be a possibility that the network would generalize to longer time delays or sequence lengths without retraining. We believe this would be an extremely interesting subject for future work and believe these types of questions serve to demonstrate the potential value of our submission as a seed for the development of future models in this direction.

---

### Official Review · Reviewer_ovPq · 2023-11-01

**Soundness:** 3 good
**Presentation:** 3 good
**Contribution:** 3 good
**Rating:** 8
**Confidence:** 4

**Summary:**

This paper presents Wave-RNN, in which the hidden states are organized in time to resemble a traveling wave.  Because the position of the wave enables one to reconstruct the time of the event that triggered it, the network can maintain precise information about what happened when in the past.   A one-layer wRNN is evaluated on some artificial tasks (e.g., psMNIST) to evaluate its ability to solve problems with long-range dependencies.  It is compared to other models, especially Identity RNN which the authors argue provides a fair comparison because it has long memory but does not exhibit waves.

**Strengths:**

This model can remember not only retain information about input to the network but also the time at which it was experienced as long as the waves persist.  Information about time is very useful.

**Weaknesses:**

The connection between this computational model and traveling waves in the cortex is extremely tenuous.  The problem is that the goal of this model is to allow information to be remembered for a long time whereas the traveling waves over the cortex (and hippocampus and striatum) are much faster.  For instance, the Siapas & Lubenov paper shows that theta travels the length of the hippocampus in on the order of 200 ms.

There is some evidence for very slow oscillations in MEC, but there is no evidence these are traveling waves.
https://doi.org/10.1101/2022.05.02.490273
https://doi.org/10.1016/j.celrep.2023.113271

On the other hand, there is extremely robust evidence for reliable sequences of firing in the brain over time scales relevant for memory.
https://doi.org/10.1038/s41467-018-07020-4
https://doi.org/10.1016/j.cub.2021.01.032
This phenomenon is often (but not always) referred to as ``time cells.'' There is not evidence that these sequence are anatomically organized, but that doesn't seem to be important for the model.

There is a critical difference between the sequences observed with time cells and the traveling waves here.  In particular, the sequences in the brain slow down as they unfold.  This is as if the wave was traveling through a fluid whose properties change systematically from one end of a channel to the other.  This does not seem to be a property of this model.  Although Figure 9 in Appendix C shows waves that travel at different speeds in different channels, the waves within each channel proceed at a constant velocity (the right panel is more complicated but certainly doesn't slow down the way the neural data do).

**Questions:**

Suppose that the velocity v in Eq. 2 changed systematically as a function of the row of the matrix.  In particular, what if v went down like 1/x?  How would that wave behave?  How would this change how the longest time horizon of the memory scales with number of weights/units?  Would this model behave better or worse on the problems in this paper than wRNN does?   Presumably worse because the temporal resolution at long delays would be poor and problems like psMNIST require precise timing information.  Could those problems be mitigated with a deep network?

---

> ### Author Response · Authors · 2023-11-23
> **Response to Reviewer ovPq (1/2)**
>
> ## Summary
> We thank the reviewer for their careful review and for their acknowledgement of the value of our contribution. To summarize our understanding of the reviewer’s comments, they are impressed with our experimental results and appear to find our contribution valuable.  The reviewer appears to be primarily concerned with the biological connection of our model and specifically the ability for traveling waves in the brain to be a viable mechanism for encoding memory. In response we have provided a few references which demonstrate the potential for traveling waves over appropriately longer time scales relevant for memory. Furthermore, we have attempted to describe our interpretation of the relation between our work and the notion of 'time cells'. Finally, we have performed the additional experiment proposed by the reviewer in their questions and included the result in our appendix with extended discussion of these points.
>
> ### Waves as a memory mechanism
> In response to the reviewer’s concern about the biological relevance of our model, and the uncertainty regarding the reality of waves as a memory mechanism in the brain, we would like to highlight two potential answers.
>
> The first is that there is indeed preliminary evidence that some traveling waves do exist over significantly longer timescales than the millisecond scale referenced by the reviewer, and that these do indeed have a relation to memory. Specifically (as noted in our introduction), "King & Wyart (2021) provide evidence that traveling waves propagating from posterior to anterior cortex serve to encode sequential stimuli in an overlapping `multiplexed' manner". In detail, they use a task of carefully constructed sequential stimuli presented over 2 seconds and show that such stimuli are decodable from EEG measurements at sequentially propagated regions of the cortex, implying a wave of activity encoding this information. Others such as Zabeh et al. (2023) also show traveling waves in frontal and parietal lobes encode memory of recent rewards. Therefore, while we certainly agree with the reviewer that certain waves (such as those which exist in the millisecond timescale of the hippocampus) are inappropriate for encoding longer term memories, we believe that there are likely a diversity of types of traveling waves in the brain and therefore this mechanism should not be discounted so quickly.
>
> The second potential answer to the reviewers concern is that the current studies of traveling wave dynamics in the brain are inherently very limited. It is simply very challenging to measure traveling waves in the brain due to the necessity for both high spatial and temporal resolution simultaneously. Therefore, while we agree that there is currently limited evidence that traveling waves may support the type of memory implied by our model, we again do not believe that this should be a reason to discount such a potential mechanism. Specifically, we believe that our model may indeed encourage further targeted study of waves. Furthermore, we note that we are not the first to suggest such a relation between waves and memory as outlined in our introduction and in the review paper (Muller et al. 2018).

---

> > ### Author Response · Authors · 2023-11-23
> > **Response to Reviewer ovPq (2/2)**
> >
> > ### Relation to Time Cells
> > The reviewer's second primary concern is how our model might relate to the well known time cells of the cortex. Specifically we would like to propose an alternative complimentary viewpoint to the reviewer's comment that  ‘the sequences in the brain slow down as they unfold’. In our understanding, from (Jacques et al. 2021 <https://arxiv.org/abs/2104.04646> & Cao et al. 2022 <https://doi.org/10.7554%2FeLife.75353>), an alternative equally valid description of the activity of time cells is as a set of basis functions which are logarithmically spaced in time. In other words, the frequency (velocity) of each neuron (wave) does not change over time, but rather different sets of neurons represent the same input over different (logarithmically spaced) time scales. This phenomena can be seen in Figure 3E of the reviewer's reference (https://doi.org/10.1016/j.cub.2021.01.032). We believe that the experiments in Figure 10 of Appendix C precisely demonstrate such a phenomenon — different speed waves effectively encode the input over different time scales. We did additionally experiment with logarithmically spaced velocities but found the presented randomly initialized velocities to perform better. We believe that the connection between time cells and the waves in our model is indeed relevant and interesting, and we thus appreciate the reviewer bringing this point to our attention. We have included a preliminary discussion of time cells in our related work, and we have updated our discussion in the appendix to reference time cells more prominently. Ultimately we believe that rather than an alternative to ideas of time cells our model may provide a complimentary view by which further discoveries may be made.
> >
> >
> > ### Questions
> > We thank the reviewer for these questions and suggestions for how the wave velocity might me modulated during wave propagation. We appreciate this highly original suggestion, it is not something we had considered before, and therefore decided to perform this additional experiment. In the updated appendix section titled 'Waves with variable velocity' we include visualizations of the hidden state of a wRNN with a hidden-to-hidden connectivity matrix initialized exactly as described by the reviewer (v scaled as 1/x where x is the row index). Indeed, when trained on sMNIST we see that the model performs roughly the same as the wRNN presented in the main text (test accuracy %97.76), and that waves appear to slow in velocity as they propagate. We agree that this might improve the longest time-horizon the model is able to handle at a sacrifice of temporal resolution, but leave precise quantification of this to future work. Ultimately we present this experiment to demonstrate that such modifications are possible in our framework, thereby increasing the benefit of the present submission to the community.

---

> > ### Comment · Reviewer_ovPq · 2023-11-23
> > **traveling waves**
> >
> > There is no advantage to arguing for traveling cortical waves measurable with EEG or ECoG electrodes.  A wave that travels over the cortex (or striatum or hippocampus) reflects the coordinated activity of *very* large numbers of neurons over very long distances (~10^5 or 10^6 larger than the density of neurons).  If we identify the units in the RNNs with individual neurons there is no reason to expect that sequences along individual neurons would be measurable at that scale in the brain.  Similarly, there's no reason to assume that the ``direction of the wavefront'' in the RNN aligns with the cortical surface, which is pretty complicated in terms of functional cell types.

---

> ### Comment · Reviewer_ovPq · 2023-11-23
> **Waves of different velocity**
>
> I read the section entitled ``Waves with Variable Velocity''.  I am not certain how one can see that the velocity is variable from Fig 19, but I appreciate the inclusion of this.  Bumping up a notch.

---

### Official Review · Reviewer_h1zv · 2023-11-03

**Soundness:** 3 good
**Presentation:** 2 fair
**Contribution:** 3 good
**Rating:** 5
**Confidence:** 4

**Summary:**

In this paper, the authors explore how patterns of wavelike activity, observed in brains, might help artificial neural networks learn and recall sequences of inputs. They discretize the 1-dimensional wave equation, finding a common structure between the discretization matrix and convolutions that they exploit to set up a simple RNN that naturally supports wavelike activity patterns.  After demonstrating that the network indeed exhibits waves, they test how such waves may facilitate sequence learning and recall in a number of different tasks by comparing the network to a very similar one initialized in a manner that does not naturally support waves.  Overall, the authors’ RNN performed impressively well across a number of sequence tasks, including a more complex permuted MNIST sequence task.

**Strengths:**

The authors overall provide a very good job in providing motivation, relevant biological background, and mathematical intuition for their network setup.  The results of their simple setup are impressive, and are generally carefully presented and analyzed, including with ablation studies.  Moreover, the authors provide a fuller characterization of the wave activity and performance of their network in the Appendix, including, transparently, their network result distributions, which is generally far too lacking in the field. Similarly, they provide their code in an anonymous repo and specify the parameters of the network, increasing the reproducibility profile of the work.

**Weaknesses:**

Note, the below concerns have resulted in a lower score, which I would be happy to increase pending the authors’ responses.

**A. Wave fields**

The wave-field comparisons, claims, and references seem a bit strained and unnecessary.  Presumably, by “wave-field,” the authors simply mean a vector field that supports wave solutions.  In any case, since this term is not oft-used in neuroscience or ML that I am aware of, a brief definition should be provided if the term is kept.  However, I am unsure that it is necessary or helpful.  That the brain supports wavelike activity is well-established, and some evidence for this is appropriately outlined by the authors.  Many computational neuroscience models support waves in a way that has been mathematically analyzed (e.g., Wilson-Cowan neural fields equations).  The authors’ discretization methodology suggests a similar connection to such analyses.  However, appealing to “physical wave fields” to relate waves and memory seems to be overly speculative and unnecessary for the simple system under study in this manuscript.  The brain is a dissipative rather than a conservative system, so that many aspects of physical wave fields may well not apply.  Moreover, the single reference the authors do make to the concept does not apply either to the brain or to their wave-RNN.  Instead, Perrard et al. 2016 describe a specific study that demonstrates that a particle-pilot wave system can still maintain memory in a very specific way that does not at all clearly apply to brains or the authors’ RNN, despite that study studying a dissipative (and chaotic) system. Instead, the readers would benefit much more from gaining an intuition as to why such wavelike activity might benefit learning and recalling sequential inputs.  Unfortunately, Fig. 1 does little to help in this vein.

However, the concept certainly is simple enough, and the authors provide a few intuitions in the manuscript that help.  I believe the manuscript would improve by removing the discussion of wave fields and instead providing / moving the intuitive explanations (e.g., the “register or ‘tape’” description on p. 20) as to how waves may help with sequential tasks to the same portion of the Introduction.

**B. Fourier analysis**

Overall, I found the wave and Fourier analysis a bit inconsistent and potentially problematic.  While I agree that the wRNNs clearly display waves when plotted directly, the mapping and analysis within the spatiotemporal Fourier domain (FD below) does not always match patterns in the regular spatiotemporal plots (RSP below).  Moreover, it’s unclear how much substance they add to the analysis results.  In more detail:

1. Constant-velocity, 1-D waves don’t need to be transformed to the FD to infer their speeds.  The slopes in the RSP correspond to their speeds.  For example, in Fig. 2 (top left), there is a wave that begins at unit 250, step ~400, that continues through to unit 0, step ~650, corresponding to a wave speed of ~1.7 units/step, far larger than the diagonal peak shown in the FD below it that would correspond to a speed of ~0.3 units/step, as indicated by the authors.

2. Similar, seemingly speed mismatches can be observed in the Appendix.  E.g., in Fig. 9 (2nd column, top), the slopes of the waves are around 0.35-0.42 units/step (close enough to likely be considered the same speed, especially as they converge in time to form a more clustered wave pulse) from what I can tell, whereas the slopes in the FD below it are ~0.3 for the diagonal (perhaps this is close enough to my rough estimate) and ~0.9, well above any observable wave speed. Perhaps there is a much faster wave that is unobservable in the RSP due to the min/max values set for the image intensity in the plot, but in that case the authors should demonstrate this.  Given (a) the potential mismatch in the speeds for the waves that can be observed, (b) the mismatch in the speeds discussed above in Fig. 2, and (c) the fact that some waves may be missed in FD (see below), I would worry about assuming this without checking.

3. As alluded to in the point above, iRNN in Fig. 2 appears to have some fast pulse bursts easily observed in the RSP that don’t show in the FD. For example, there is a very fast wave observable in the RSP in units ~175-180, time steps 0-350.  Note, the resolution is poor, but zooming in and scrolling to where the wave begins around unit 175, step 0 makes it clear.  If one scrolls vertically such that the bottom of the wave at step 0 is just barely unobservable, then one can see the wave rapidly come into view and continue downwards.  Similarly some short-lasting, slower pulses in units near 190, steps 0-350 are observable in the RSP.  None of these appear in the FD.  Note, this would not take away from the claim that wRNNs facilitate wave activity much more than iRNNs do, but rather that some small amounts—likely insufficient amounts for facilitating sequence learning—of wave activity might still arise in iRNNs.  If the authors believe these wavelike activities are aberrant, it would be helpful for them to explain why so.

4. I looked over the original algorithm the authors used (in Section III of “Recognition and Velocity Computation of Large Moving Objects in Images”—RVC paper below—which I would recommend for the authors to cite), and I wonder if an error in the initial calibration steps (steps 1 & 2) occurred that might explain the speed disparities observed between the RSPs and FDs.

5. There do seem to be some different wave speeds—e.g., in Fig. 9, there appear to be fast and narrow excitatory waves overlapping with slow and broad inhibitory waves. But given that each channel has its own wave speed parameter $\nu$, it isn’t clear why a single channel would support multiple wave speeds.  This should be explored in greater depth, and if obvious examples of sufficiently different speeds of excitatory waves are known (putatively Fig. 9, 2nd column), these should be clearly shown and carefully described and analyzed.

6. Is there cross-talk across the channels?  If so, have the authors examined the images of the hidden units (with dimensions __hidden units__ x __channels__) for evidence of cross-channel waves?  If so, perhaps this is one reason for multiple wave speeds to exist per channel?

7. Overall, it is unclear overall what FT adds to the detection of 1-D waves.  If there are such waves, we should be able to observe them directly in the RSPs.  In skimming over the RVC paper, it seems like it would be most useful in determining velocities of 2-D objects and perhaps wave pulses.  That suggests that one place the FD analysis might be useful is if there are cross-channel waves as I mention above.  If so, the waves should still be observable in the images (and I would encourage such images be shown), but might be more easily characterized following the marginalization decomposition procedure described in the original algorithm in Section III of the RVC paper.  Note, the FDs might also facilitate the detection of multiple wave speeds in the network, as potentially shown in Fig. 9.  However, in that case it would seem they should only appear in Fig. 9, and if the speeds are otherwise verified.

8. The authors mention they re-sorted the iRNN units to look for otherwise hidden waves.  This seems highly problematic.  If there are waves, then re-sorting can destroy them, and if there is only random activity then re-sorting can cause them to look like waves.

**C. Mechanisms**
Finally, while the results are overall impressive, and hypotheses made regarding the underlying mechanisms for the performance levels of the network, there is too little analysis of the these mechanisms.  While the ablation study is important and helpful, much more could be done to characterize the relationship between wavelike activity and network performance.

**D. Minor**
1. Fig. 2: Both plots on the right have the leftmost y-axis digits obscured
2. Fig. 9, top, plots appear to have their x- and y- labels transposed (or else the lower FD plots and those in Fig. 2 have theirs transposed.
3. Fig. 15 needs axis labels

**Questions:**

Please see **Weaknesses**

---

> ### Author Response · Authors · 2023-11-23
> **Response to Reviewer h1zv (1/3)**
>
> ## Summary
> We thank Reviewer h1zv for their thorough review and constructive commentary, we believe it has truly helped us improve our manuscript and we appreciate the reviewer's time. Below we summarize the reviewer's comments and our responses:
>
> First, (A) it appears the reviewer is primarily concerned by the direct description of neural activity in terms of physical wave dynamics and would prefer an alternative description of the hidden state as a 'register'.
> - In response we have altered our introduction to include the mechanistic 'register' intuition, and altered our phrasing to highlight the physical wave comparison is an imperfect analogy.
>
> Secondly, (B) they were concerned with apparent inconsistency between the wave velocity estimates obtained from the regular spatio-temporal activity plots and the Fourier domain plots, leading them to question the value of the Fourier analysis.
> - In response we have identified the issue (an initial incorrect calibration based on sampling rate) and corrected the axes of our Fourier analysis plots. We have additionally included a new figure in the appendix (Figure 12) validating our calibration with synthetic data.
>
> In the sections below we adress these concerns in more detail. We note that in our updated submission we have highlighted all text changes in blue to allow for quicker evaluation.
>
>
> ### Weakness A - Wave fields
> The reviewer is correct in understanding the name 'wave-field' is simply a reference to the hidden state of the network which supports and encourages wave solutions. This is a term derived from physics to describe "the extended area or space taken up by a wave" (wiktionary). We decided to use this name in order to distinguish this type of hidden state from the baseline RNN hidden state which has no notion of spatial organization and no bias towards traveling wave dynamics. Our initial motivation for appealing to physical wave dynamics was in reference to the review paper (Muller et al. 2018) in which they make an analogy with the pilot-wave work of (Perrard et al. 2016). We include a quote below from that review  which we believe may directly address some of the reviewer's concerns related to its inclusion:
> > A recent study examined the dynamics of a silicone droplet bouncing on a vibrating oil bath (Perrard et al. 2016). The vibrations of this medium propel the droplet upward and cause it to bounce continually, creating a pattern of successive impacts across the surface (Fig. 6a). With each bounce, the droplet creates a set of waves in the oil bath. In our analogy, these droplet waves may be similar to those created in the sensory cortex by the impact of afferent, stimulus-driven action potentials. The waves evoked by each impact of the droplet then create a pattern across the oil bath that leaves a trace of the path of the droplet. With sufficient knowledge of the dynamic properties of the medium, one could decode the history of the path of the droplet simply from the observed spatial pattern at each point in time. Furthermore, temporal reversibility is preserved for waves even in chaotic systems; this property, along with a clever manipulation of the experimental apparatus, can be used to cause the droplet to exactly reverse its trajectory (Fig. 6a). It is argued that this last manipulation shows that the global wave field stores information about the trajectory of the droplet with sufficient precision that it can be read back out. While the authors did not have neuroscience in mind and, indeed, note that they have not yet considered practical implementations, they posit that this system implements the basic elements for general computations that could find application in future work.
>
> We appreciate that the reviewer has highlighted that our initial emphasis on this analogy may have been distracting to readers. We have therefore followed the reviewer's suggestion and rephrased our introduction to highlight this is an imperfect analogy while including the more intuitive `register' description earlier on. We have additionally updated Figure 1 to match the wRNN dynamics more closely and resemble physical wave dynamics less.

---

> > ### Author Response · Authors · 2023-11-23
> > **Response to Reviewer h1zv (2/3)**
> >
> > ### Weakness B - Fourier Analysis
> > We thank the reviewer for highlighting the inconsistency of wave velocity estimates from our Fourier analysis and regular spatio-temporal analysis. Thanks to the reviewer's comments, we have indeed located a miscalibration of our calculation of the axes labels in the original submission. Specifically, in the prior submission, we did not consider the sampling rate, and therefore did not have correct bounds on our axes. In response we have updated all Fourier analysis plots in our submission, and encourage the reviewer to re-validate their consistency with the regular spatio-temporal velocity estimates. To facilitate this validation, we have additionally included a new set of calibration experiments in the appendix (Figure 12), where we have computed the 2D Fourier transform of synthetic data with known velocities and observed slopes of peak magnitude in the Fourier domain which match the known velocities.
> >
> > In response to the reviewer's concern that the Fourier analysis may be superfluous, while we agree this may be the case for the example in Figure 2 where waves are very clear in the regular domain, we found the Fourier domain plots to be highly beneficial in developement of the model and in studying baselines where the existence of waves is much less clear cut. This is exemplified by the prior work on traveling waves (Davis et al., 2021) in which the authors strongly rely on such analysis to validate which aspects of model architecture yield waves. Ultimately we believe this analysis is helpful both as a second point of validation for the existence/lack of traveling waves in the wRNN/iRNN, as well as a demonstration of a tool for researchers who may be interested in continuing this line of work. We further believe with the corrected calibration there is little negative side effect to including this extra analysis in the paper, and therefore have left it in our updated submission.
> >
> > To address a few of the reviewer's other more direct questions:
> > >  Is there cross-talk across the channels? If so, have the authors examined the images of the hidden units (with dimensions hidden units x channels) for evidence of cross-channel waves? If so, perhaps this is one reason for multiple wave speeds to exist per channel?
> >
> > Yes, the convolutional kernel $\mathbf{u}$ is fully parametrized and thus allows for the potential for activity to propagate between channels if weights are optimized that way. In the updated submission, we have included a new figure (Figure 9) which visualizes activity for all channels of the wRNN. However, since there is no implicit ordering of the channels, it is similar to the situation with the iRNN where it will be difficult to directly identify cross-channel waves since they may move between arbitrary channel indexes at each time-point (see comment below). In practice however, we agree with the reviewer that this is likely the reason multiple wave speeds can exist in a single channel, since each channel does have its own velocity parameter which only supports a single velocity independently.
> >
> >
> > >  The authors mention they re-sorted the iRNN units to look for otherwise hidden waves. This seems highly problematic. If there are waves, then re-sorting can destroy them, and if there is only random activity then re-sorting can cause them to look like waves.
> >
> > We would like to clarify what appears to be a potential confusion by the reviewer in this comment. In the iRNN there is no notion of spatial locality, so it does not make sense to speak of waves in the network hidden state without any sorting. As another way to understand this, in a network without spatial biases (such as the iRNN) ‘waves’, if they do exist, would be equally likely to propagate between any two neurons in the hidden state over time (not just those which are arbitrarily adjacent to one another in our hidden state vector). Therefore, in order to identify if there is any smooth continuous flow of activity between any neurons (mathematically equivalent to a wave), it makes sense to sort activity according to maximal ‘onset time’. Indeed, due to this sorting we do see the short wave-bursts the reviewer has identified, however they simply do not contain enough power to be visible in the fourier domain, and additionally do not persist long enough to be useful as a form of memory. We do appreciate that this sorting may raise concern in readers and thank the reviewer for bringing it up. To address this, in the updated paper appendix (Figure 11) we have included the unsorted iRNN activations where significantly less structure is present, as is expected.

---

> > > ### Author Response · Authors · 2023-11-23
> > > **Response to Reviewer h1zv (3/3)**
> > >
> > > ### Weakness C -- Mechanism
> > > We appreciate the reviewer’s concern that the overall mechanism could be better explained. In the updated version of the paper we have updated our introduction and initial figure to describe the intuition of this mechanism more clearly. We hope this partially alleviates the reviewers concerns, however we certainly agree that more could be done to understand the 'impressive' performance of the wave-RNN. Overall, we believe mechanistic understanding is often proceeded by empirical findings which we believe contain significant value to the community on their own. We therefore present this work as a working model of wave-based memory, opening the door to future work which may improve mechanistic understanding.
> > >
> > > ### Minor Weaknesses D
> > > We thank the reviewer for highlighting the graphical issues with some of our plots. We have fixed these in the updated revision.

---

### Author Response · Authors · 2023-11-23

### Overall Response
We thank all parties involved for their careful evaluation of our work, for their creative suggestions for improvement, and for ultimately helping to improve the overall quality of our submission.

We appreciate the positive views expressed by reviewers that our work is **rigorous and well motivated** (6SWP, h1zv), **novel** (2PWQ), **timely and important** (6SWP), with **impressive and generally carefully presented** results (h1zv).

In our individual responses to reviewers below we have addressed their precise concerns directly, and provided an updated submission pdf. In our updated submission, we have highlighted in blue portions of the text which have been modified to allow for quick validation by the reviewers and AC. To summarize the main changes we have made to the submission:
- we have improved the provided intuition in our introduction in response to the helpful comments from reviewers h1zv and 2PWQ, and modified Figure 1 to better support this intuition.
- we have modified the description of our model and notation to make the multi-channel nature of our hidden state more clear.
- we have fixed calibration of our Fourier analysis, and included evidence of correct calibration in the appendix.
- we have included additional visualizations in the appendix to address reviewer concerns, such as: visualization of wRNN hidden state for all channels, visualization of the unsorted iRNN hidden state, and visualizations of waves with slowly decreasing velocity.

---

### Meta-Review · Area_Chair_rXHb · 2023-12-14

**Metareview:**

This work presents a recurrent network inspired by the biological observation of traveling waves across cortex (as has observed in EEG and fMRI). To this end, the so-called wave RNN (wRNN) organizes unites to activate sequentially over time, ideally preserving temporal information. The memory retention seems to hold, as the authors show on a number of examples, in particular highlighting the performance-per-parameter ratio.  The reviewers raised a number of points related to the motivation, model analysis, biological relationship, and clarity of exposition in the paper. The authors addressed a number of these, however the model analysis (i.e., why is this model with memory a gain over other architectures more generally) was less addressed. This is a borderline case, however given the generally positive responses on the novelty of the approach and results, I recommend accepting this work.


As an aside to the authors: There is the additional aspect of memory, in general, being studied in the echo-state and liquid-state machine literature. While slightly different in the sense of having random weights, the analysis may be relevant to the specific points made by the authors with respect to memory capacity and timing.

**Justification For Why Not Higher Score:**

The many concerns on the motivation and understanding/analysis of the model drove the score down.

**Justification For Why Not Lower Score:**

Primarily, the novelty of the approach and diversity of model validation and results drove the score up.

---

### Decision · Program_Chairs · 2024-01-16

Accept (poster)